# Kinesin Kif2C in regulation of DNA double strand break dynamics and repair

**Songli Zhu[1†], Mohammadjavad Paydar[2†], Feifei Wang[1,3], Yanqiu Li[1], Ling Wang[1], Benoit Barrette[2], Tadayoshi Bessho[4], Benjamin H Kwok[2*], Aimin Peng[1*]**

[1]Department of Oral Biology, College of Dentistry, University of Nebraska Medical Center, Omaha, United States; [2]Institute for Research in Immunology and Cancer (IRIC), Département de médecine, Université de Montréal, Montréal, Canada; [3]Institute of Physical Science and Information Technology, Anhui University, Hefei, China; [4]Eppley Institute for Research in Cancer and Allied Diseases, Fred & Pamela Buffett Cancer Center, University of Nebraska Medical Center, Omaha, United States

**Abstract** DNA double strand breaks (DSBs) have detrimental effects on cell survival and genomic stability, and are related to cancer and other human diseases. In this study, we identified microtubule-depolymerizing kinesin Kif2C as a protein associated with DSB-mimicking DNA templates and known DSB repair proteins in *Xenopus* egg extracts and mammalian cells. The recruitment of Kif2C to DNA damage sites was dependent on both PARP and ATM activities. Kif2C knockdown or knockout led to accumulation of endogenous DNA damage, DNA damage hypersensitivity, and reduced DSB repair via both NHEJ and HR. Interestingly, Kif2C depletion, or inhibition of its microtubule depolymerase activity, reduced the mobility of DSBs, impaired the formation of DNA damage foci, and decreased the occurrence of foci fusion and resolution. Taken together, our study established Kif2C as a new player of the DNA damage response, and presented a new mechanism that governs DSB dynamics and repair.

**\*For correspondence:**
benjamin.kwok@gmail.com (BHK);
aimin.peng@unmc.edu (AP)

[†]These authors contributed equally to this work

**Competing interests:** The authors declare that no competing interests exist.

## Introduction

DNA damage is frequently induced by both endogenous metabolic products and exogenous geno-toxic agents. Upon DNA damage, the cell promptly activates the cellular DNA damage response (DDR), a surveillance mechanism that leads to DNA repair, cell cycle arrest (checkpoint), and apoptosis (*Li and Zou, 2005*; *Lou and Chen, 2005*; *Zhou and Elledge, 2000*). Among all types of DNA damage, DNA double strand break (DSB) is of great toxicity and deleterious consequences. It is therefore crucial for cells to efficiently repair DSBs, whereas defects in DSB repair have been linked to cancer, immunodeficiency, neurological diseases, and aging (*Jalal et al., 2011*; *Liang et al., 2009*; *Sancar et al., 2004*).

The cell employs two major evolutionarily-conserved mechanisms, non-homologous end joining (NHEJ) and homologous recombination (HR) to repair DNA DSBs (*Goodarzi and Jeggo, 2013*; *Sancar et al., 2004*). HR restores the broken DNA strands using an intact strand as template, and is available in S and G2 phases after replication of chromatin DNA (*Jasin and Rothstein, 2013*). By comparison, NHEJ directly re-ligates the two broken ends of a DSB, and is accessible throughout the entire interphase (*Davis and Chen, 2013*; *Lieber, 2010*). In addition to these core pathways of DSB repair, the spatiotemporal regulation of DSBs has emerged as a new aspect of DNA repair (*Amitai et al., 2017*; *Chuang et al., 2006*; *Chung et al., 2015*; *Hauer and Gasser, 2017*; *Krawczyk et al., 2012*; *Lemaître and Soutoglou, 2015*; *Levi et al., 2005*; *Lottersberger et al., 2015*; *Marcomini et al., 2018*; *Marnef and Legube, 2017*; *Miné-Hattab and Rothstein, 2013*; *Neumaier et al., 2012*; *Schrank et al., 2018*). Potentially, the physical mobility of DSBs mediates

**eLife digest** DNA can be damaged in many ways, and a double strand break is one of the most dangerous. This occurs when both strands of the double helix snap at the same time, leaving two broken ends. When cells detect this kind of damage, they race to get it fixed as quickly as possible. Fixing these double strand breaks is thought to involve the broken ends being moved to 'repair centers' in the nucleus of the cell, but it was unclear how the broken ends were moved.

One possibility was that the cells transport the broken ends along protein filaments called microtubules. Cells can assemble these track-like filaments on-demand to carry cargo attached to molecular motors called kinesins. However, this type of transport happens outside of the cell's nucleus, and while there are different kinesin proteins localized inside the nucleus, their roles are largely unknown.

In an effort to understand how broken DNA ends are repaired, Zhu, Paydar et al. conducted experiments that simulated double strand breaks and examined the proteins that responded. The first set of experiments involved mixing cut pieces of DNA with extracts taken from frog eggs or human cells. Zhu, Paydar et al. found that one kinesin called Kif2C stuck to the DNA fragments, and attached to many proteins known to play a role in DNA damage repair. Kif2C had previously been shown to help separate the chromosomes during cell division. To find out more about its potential role in DNA repair, Zhu, Paydar et al. then used a laser to create breaks in the DNA of living human cells and tracked Kif2C movement. The kinesin arrived within 60 seconds of the DNA damage and appeared to transport the cut DNA ends to 'repair centers'. Getting rid of Kif2C, or blocking its activity, had dire effects on the cells' abilities to mobilize and repair breaks to its DNA. Without the molecular motor, fewer double strand breaks were repaired, and so DNA damage started to build up.

Defects in double strand break repair happen in many human diseases, including cancer. Many cancer treatments damage the DNA of cancer cells, sometimes in combination with drugs that stop cells from building and using their microtubule transport systems. Understanding the new role of Kif2C in DNA damage repair could therefore help optimize these treatment combinations.

the sub-nuclear organization and positioning of DSBs to facilitate DNA repair. However, the precise mechanisms which propel and regulate DSB mobility remain largely obscure.

Microtubules (MTs) are composed of α/β tubulin dimers, and responsible for a variety of cell movements, including the intracellular transport of various vesicles and organelles, and separation of chromosomes in mitosis (*Dogterom et al., 2005*; *Forth and Kapoor, 2017*; *Maizels and Gerlitz, 2015*). For example, cargos, including proteins, nucleic acids and organelles, can be moved along MTs by the action of motor proteins which utilize ATP hydrolysis to produce force and movement (*Dogterom et al., 2005*; *Forth and Kapoor, 2017*; *Maizels and Gerlitz, 2015*). A major group of molecular motors involved in intracellular transport are kinesins named Kif (kinesin superfamily protein). There are several dozen Kifs in mammalian cells to constitute at least 14 kinesin families (*Hirokawa et al., 2009*; *Lawrence et al., 2004*). Unlike most kinesins, Kif2C, also known as Mitotic Centromere Associated Kinesin or MCAK, and other members of the kinesin-13 family do not utilize their ATPase activities to transport cargos, but rather to depolymerize MTs by disassembling tubulin subunits at polymer ends (*Desai et al., 1999*; *Hunter et al., 2003*; *Walczak et al., 2013*; *Wordeman and Mitchison, 1995*). During cell division, Kif2C regulates MT dynamics and ensures the proper attachment of MTs to kinetochores, and thereby directing the positioning and movement of chromosomes (*Ganem et al., 2005*; *Kline-Smith et al., 2004*; *Manning et al., 2007*). In this study we identify and characterize Kif2C as a new factor involved in DSB repair; Kif2C is required for efficient DSB repair via both HR and NHEJ; and interestingly, Kif2C facilitates DSB mobility and modulates the formation, fusion, and resolution of DNA damage foci.

## Results

### Kif2C associates with DSB-mimicking substrates and DNA repair proteins

As described in our previous study (*Zhu et al., 2017*), we utilized DNA DSB-mimicking dA-dT oligonucleotides to isolate potential DNA damage-associated proteins in *Xenopus* egg extract, a cell-free system well-defined for studying DNA damage repair and signaling (*Guo et al., 1999*; *Lupardus et al., 2007*). Along with Ku70, PARP1, RPA, and many other factors known to be involved in DSB repair, Kif2C was proteomically identified as a co-precipitated protein of dA-dT. We confirmed, in both *Xenopus* egg extracts and human cell lysates, that Kif2C bound another, and longer, DSB-mimicking template (*Figure 1A and B*). We then supplemented in the extract either uncut, circular plasmid DNA, or linearized plasmid DNA with free DSB ends. Interestingly, Kif2C associated specifically with the cut plasmid DNA (*Figure 1C*), further indicating that Kif2C is a DSB-associated protein.

Next, we carried out proteomic analysis to identify proteins that were associated with Kif2C. This effort recovered a number of well-established DNA damage response proteins, including Ku70/Ku80, a DSB end binding complex, H2AX, a histone variant that is phosphorylated in chromatin regions flanking DSBs, and PARP1, an early responder of various DNA lesions (*Figure 1D*). The association of Kif2C with these DNA damage factors was subsequently confirmed using both pull-down and immunoprecipitation (*Figure 1E*, *Figure 1—figure supplement 1A and B*). Treatment with DNase did not disrupt the protein association (*Figure 1—figure supplement 1C*), suggesting that it was not mediated by DNA. It has been revealed that the catalytic function of Kif2C is mediated through a motor domain located in the middle region of the protein (*Ems-McClung et al., 2007*; *Maney et al., 2001*). Interestingly, both this middle region and the N-terminus of Kif2C exhibited appreciable levels of associations with DNA repair proteins (*Figure 1—figure supplement 1D and E*), suggesting the involvement of these motifs in DNA repair.

### Kif2C undergoes two-stage recruitment to DNA damage sites

The identification of Kif2C as a potential DSB-associated protein was largely unexpected, given that MT assembly is viewed as a cytoplasmic event, except in mitosis after nuclear envelop breakdown. On the other hand, Kif2C is primarily localized to the nucleus in interphase, but the function of Kif2C in intra-nuclear events is unknown. We showed in HeLa cells that Kif2C was recruited to DNA damage sites induced by laser micro-irradiation (*Figure 2A* and *Video 1*). Kif2C was enriched at laser-irradiated sites within 1 min, indicating it as an early responder to DNA damage (*Figure 2A and B*). We confirmed subsequently that Kif2C co-localized with γ-H2AX foci induced by ionized radiation (IR, *Figure 2C*); Kif2C foci co-localized and co-migrated with 53BP1 foci in cells treated with etoposide (*Figure 2—figure supplement 1* & *Video 2*). Immunofluorescent staining analysis of endogenous Kif2C revealed consistent pattern of co-localization with IR-induced γ-H2AX foci (*Figure 2—figure supplement 2A and B*).

The fast recruitment of Kif2C to DNA damage sites prompted us to examine its dependence on PARP1-mediated PARylation, which undergoes rapid induction (<1 min) and removal (5–10 min). Interestingly, PARP inhibition disrupted the initial recruitment of Kif2C to laser-induced DNA damage sites; in the presence of a PARP inhibitor (PARPi), Kif2C slowly accumulated at DNA damage sites at about 10 min (*Figure 2D and E*). By contrast, the sustained, but not the initial, recruitment of Kif2C was dependent on ATM (*Figure 2D and E*).

To reveal additional molecular insights into the DNA damage recruitment of Kif2C, we generated multiple truncated segments of Kif2C, and examined their localization in laser-irradiated cells. Interestingly, the N-terminus of Kif2C exhibits efficient recruitment to DNA damage sites; the middle region of Kif2C containing the catalytic motif was very weakly enriched at DNA damage sites; and the C-terminus of Kif2C did not accumulate at DNA damage sites (*Figure 2F and G*). Consistent with the strong recruitment of Kif2C N-terminus to DNA damage sites, a Kif2C mutant deleted of the N-terminus was deficient in the DNA damage recruitment (*Figure 2—figure supplement 2C and D*). To identify minimal elements within the N-terminus that mediate DNA damage recruitment, we generated a series of truncation mutants within the N-terminus (*Figure 2H*). Interestingly, the efficient recruitment of Kif2C N-terminus depended on both a short five amino acid (aa 86–90) and

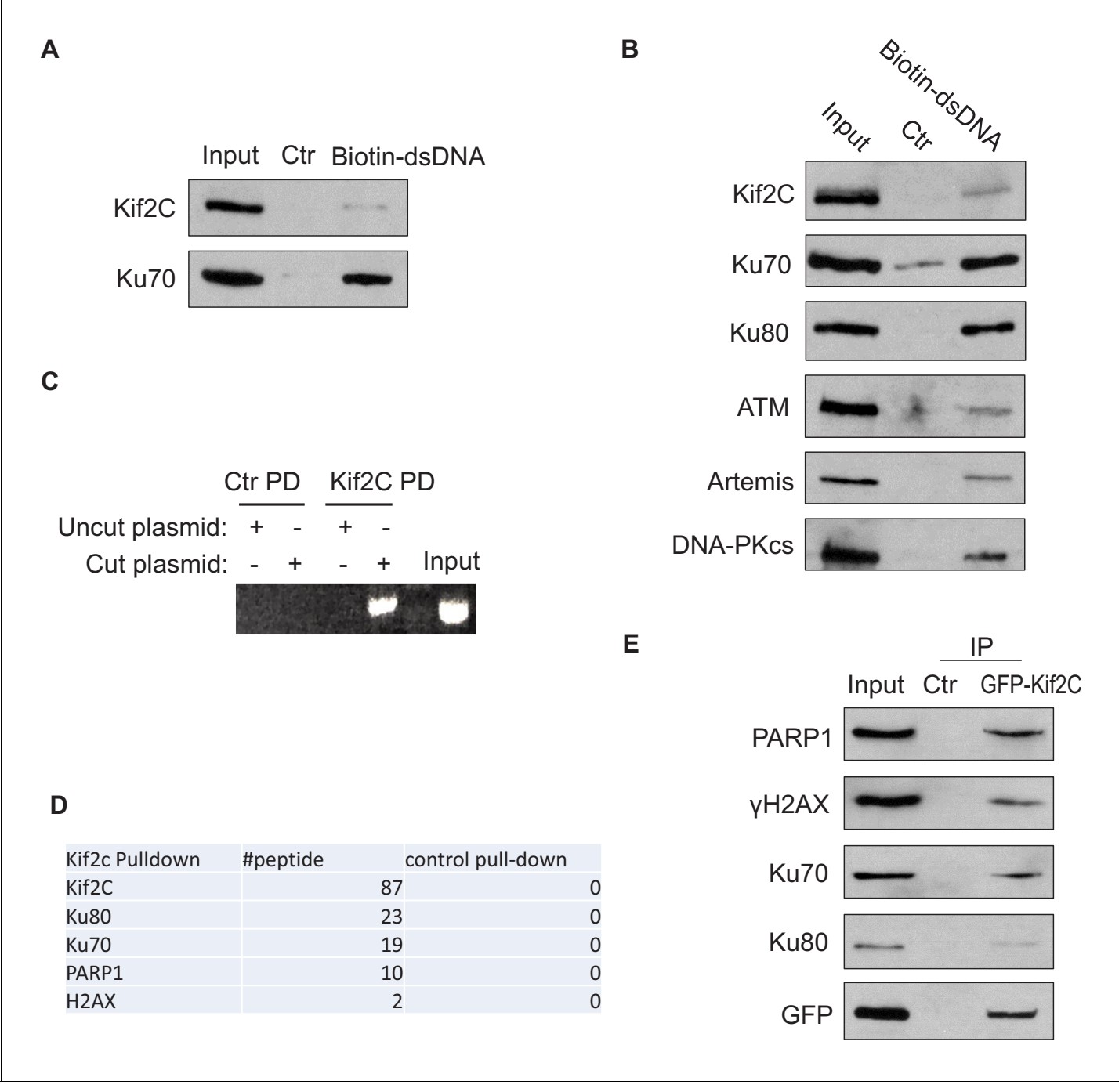

**Figure 1.** Kif2C associates with DNA double strands breaks and DNA repair proteins. (**A**) Beads conjugated with a biotin-double stranded DNA fragment (dsDNA, 500 bp, as described in Materials and methods—DNA binding assay) were incubated in *Xenopus* egg extracts for 30 min, re-isolated, and resolved by SDS-PAGE. The input, control pull-down (with blank beads), and biotin-dsDNA pull-down were analyzed by immunoblotting. (**B**) Beads conjugated with biotin-dsDNA (as in panel A) were incubated in HeLa cell lysates for 30 min, re-isolated, and resolved by SDS-PAGE. The input, control pull-down (with blank beads), and biotin-dsDNA pull-down were analyzed by immunoblotting. (**C**) *Xenopus* Kif2C was expressed with MBP-tag, and purified on amylose beads. As described in Materials and methods—pull-down assay, MBP-Kif2C or control (blank) beads were incubated in *Xenopus* egg extracts supplemented with cut or uncut plasmid, re-isolated, and analyzed by PCR and agarose gel electrophoresis/ethidium bromide staining. (**D**) As described in Materials and methods—pull-down assay, human Kif2C was expressed with MBP-tag and purified on amylose beads. MBP-Kif2C or control (blank) beads were incubated in the lysates of doxorubicin-treated HeLa cells. Pull-down samples were analyzed by mass spectrometry. The identified DNA repair proteins and numbers of peptides are shown. (**E**) GFP-Kif2C was expressed in HeLa cells with

*Figure 1 continued on next page*

*Figure 1 continued*
doxorubicin-treatment. Immunoprecipitation (IP) was performed using anti-GFP or control (blank) beads. 10% input, control and GFP IP samples were analyzed by immunoblotting.
The online version of this article includes the following figure supplement(s) for figure 1:

**Figure supplement 1.** Kif2C associates with DNA repair proteins.

the neck domain (*Figure 2H*). The neck domain of Kif2C was shown to play a role in MT depolymerization (*Maney et al., 2001*), hence, our study indicates an additional function of this domain in the DNA damage recruitment of Kif2C. By comparison, the aa 86–90 region lies outside of the minimal functional domain of Kif2C's MT-depolymerizing activity and is not associated with any known mitotic functions of Kif2C. Interestingly, both of these motifs are important for Kif2C recruitment to DNA damage sites as full-length Kif2C deleted of either one exhibited reduced recruitment to the sites of laser cut (*Figure 2I and J*). Consistent with the recruitment deficiency, these mutants also exhibited reduced association with DNA repair proteins (*Figure 2—figure supplement 3*).

## Kif2C depletion or inhibition leads to accumulation of endogenous DNA damage

As we revealed the recruitment of Kif2C to DNA damage, and the association of Kif2C with DSB templates and repair factors, we set out to investigate the function of Kif2C in the DDR. Interestingly, Kif2C knockdown in HeLa cells led to γ-H2AX induction (*Figure 3A*). The induction of γ-H2AX was also detected in U2OS cells deleted of Kif2C using CRISPR-Cas9-mediated gene editing (*Figure 3B*). Moreover, cells depleted of Kif2C exhibited increased foci formation of γ-H2AX and 53BP1 (*Figure 3—figure supplement 1*). These lines of evidence suggested that Kif2C plays a role in DNA repair, and its removal caused accumulation of endogenous DNA damage. Consistent with this hypothesis, accumulation of DNA breaks in Kif2C knockout cells was shown using a single cell electrophoresis (comet) assay (*Figure 3C*). As expected, the re-expression of RNAi-resistant Kif2C rescued γ-H2AX induction (*Figure 3D*). By comparison, a G495A Kif2C mutant defective in ATP hydrolysis and MT depolymerization, as characterized previously (*Wang et al., 2012b*), was ineffective in suppressing endogenous DNA damage caused by Kif2C depletion (*Figure 3D* and *Figure 3—figure supplement 1*), suggesting that the ATPase activity of Kif2C is required for its function in DNA repair. Previously reported structural insights into the enzymatic action of Kif2C revealed that tubulin-binding, in addition to ATPase, is required for MT depolymerization (*Ritter et al., 2016*; *Wang et al., 2017*). For example, a β5 motif with in the motor domain of Kif2C recognizes the distal end of β-tubulin, and R420S, a specific mutation in this motif disrupted tubulin-binding and MT depolymerization (*Ritter et al., 2016*). Like G495A, R420S mutant failed to rescue the accumulation of γ-H2AX in Kif2C knockout cells (*Figure 3E*), despite that these mutants were expressed at similar levels as WT (*Figure 3D and E*), and exhibited nuclear localization and DNA damage recruitment (*Figure 3—figure supplement 2A*). Kif2C depletion or mutation did not cause significant disruption of cell cycle progression (*Figure 3—figure supplement 2B*). Interestingly, Kif2C depletion did not additively enhance the induction of γ-H2AX in cells pre-treated with nocodazole, an inhibitor of MT assembly, suggesting that Kif2C functions in DNA repair in the context of MT assembly (*Figure 3—figure supplement 3A*). Moreover, the Δ86–90 Kif2C mutant deficient in DNA damage recruitment was incapable of suppressing endogenous DNA damage in Kif2C KO cells (*Figure 3F*), indicating that the DNA damage recruitment of Kif2C is required for the prevention of DSB accumulation. Together, these findings suggested that both the DNA damage recruitment of Kif2C and its catalytic activity are likely involved in the DDR.

The Kwok laboratory previously identified DHTP (((Z)−2-(4-((5-(4-chlorophenyl)−6-(iso-propoxycarbonyl)−7-methyl-3-oxo-3,5-dihydro-2H-thiazolo[3,2-a]pyrimidin-2-ylidene)methyl)phe-noxy)acetic acid)) as an allosteric inhibitor of Kif2C (*Talje et al., 2014*). Interestingly, DHTP treatment in HeLa cells phenocopied Kif2C depletion and caused γ-H2AX accumulation (*Figure 3G*). Although Kif2C also plays a role in mitosis (*Manning et al., 2007*), DHTP induced γ-H2AX accumulation efficiently in thymidine-arrested interphase cells (*Figure 3H*), indicating that mitotic defects were not the primary cause of DNA damage.

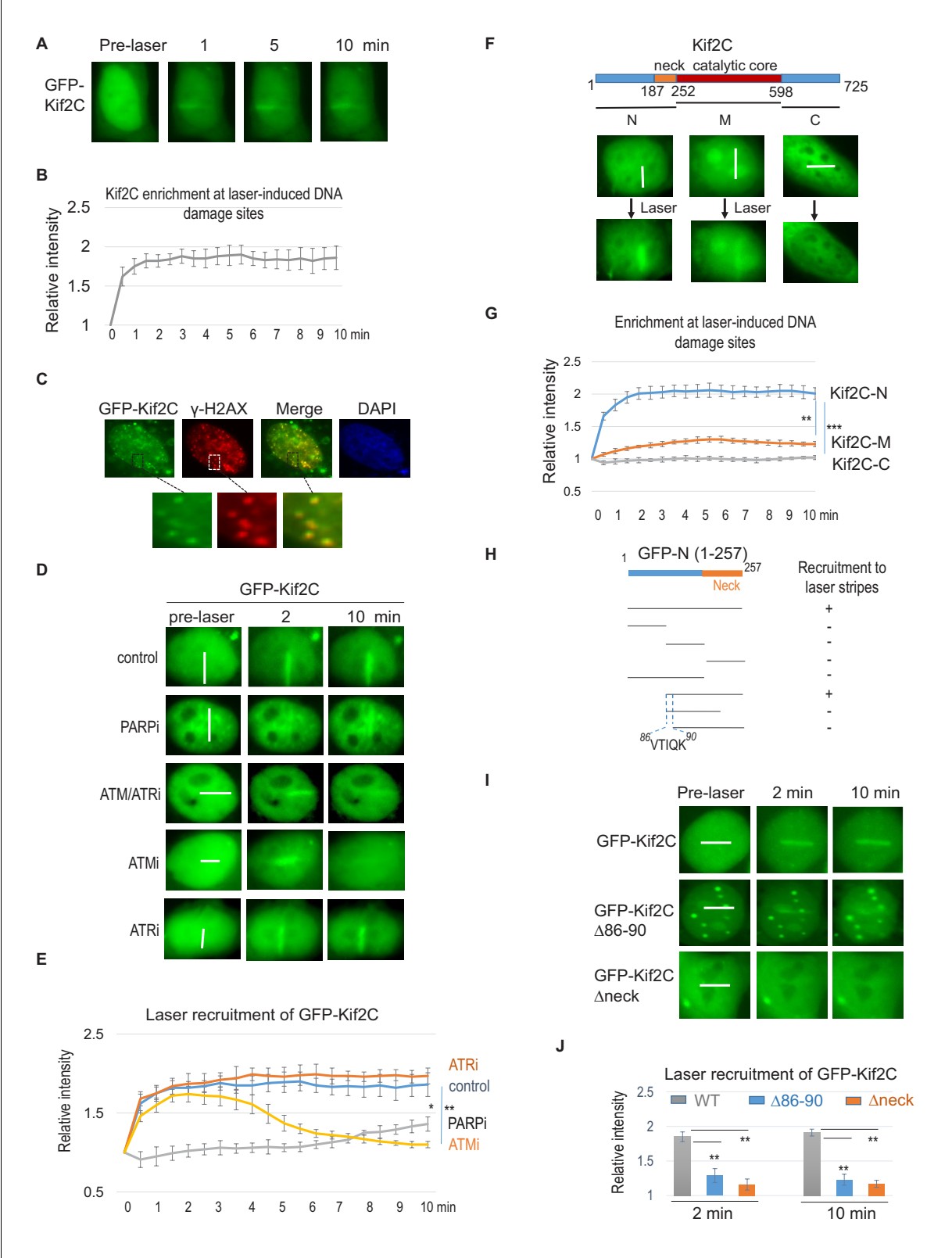

**Figure 2.** Kif2C is recruited to DNA damage sites in a two-stage manner. (**A**) HeLa cells expressing GFP-Kif2C were subjected to laser micro-irradiation as described in Materials and methods—immunofluorescence and imaging. The fluorescent signal of GFP is shown at the indicated time points. (**B**) The intensity of the GFP signal at laser-cut sites was normalized to that outside of the laser-cut sites for the relative enrichment of GFP-Kif2C. The mean values and standard deviations are shown (quantification shown in 5 cells, consistent pattern observed in >10 cells and >3 independent experiments).
*Figure 2 continued on next page*

*Figure 2 continued*

(**C**) HeLa cells expressing GFP-Kif2C were treated with 10 Gy IR, the fluorescent signal of GFP and immunofluorescent signal of γ-H2AX are shown. Pre-extraction was performed by placing the dish on ice for 5 min with 0.1% Triton X-100 in 10 mm HEPES (pH 7.4), 2 mm MgCl2, 100 mm KCl, and 1 mm EDTA. (**D**) GFP-Kif2C was expressed in HeLa cells. Prior to laser-micro-irradiation, these cells were pre-treated with PARPi (olaparib, 10 μM), ATM/ATRi (caffeine, 2 mM), ATMi (Ku55933, 5 μM), or ATRi (Ve-821, 10 μM), as indicated. The localization of GFP-Kif2C at the indicated time points is shown. The white lines mark the regions of laser micro-irradiation. Consistent results were observed in >10 cells for each treatment. (**E**) The DNA damage recruitment of Kif2C was examined as in panel D. The intensity of the GFP signal at laser-cut sites was normalized to that outside of the laser-cut sites for the relative enrichment of GFP-Kif2C. The mean values and standard deviations are shown (N = 5). ATM/ATRi showed similar kinetics as ATMi. P values were determined by two-tailed Student's t-test (*<0.05, **<0.01, ***<0.001). (**F**) The N-terminus, middle segment (**M**), and C-terminus of MCAK was expressed with a GFP tag to examine their localization in laser-treated HeLa cells. The white lines mark the regions of laser micro-irradiation. Consistent results were observed in >10 cells for each segment. (**G**) The DNA damage recruitment of Kif2C-N, M, and C was examined as in panel F. The intensity of the GFP signal at laser-cut sites was normalized to that outside of the laser-cut sites for the relative enrichment. The mean values and standard deviations are shown (N = 5). (**H**) A series of truncation mutants were generated from the N-terminus of Kif2C. These mutants, tagged with GFP, were analyzed for recruitment to laser-stripes 10 min after the treatment. The result of positive or negative recruitment was determined by consistent results in >10 cells. (**I**) GFP-Kif2C deleted of aa 86–90 or neck-motif was expressed in HeLa cells which were micro-irradiated by laser (as marked by white lines). Both the aa 86–90 and neck-motif of Kif2C are required for the efficient recruitment of Kif2C to laser stripes. The white line marks the path of laser. (**J**) The recruitment to laser stripes, as in panel I, was quantified for Kif2C (WT, or deleted of the aa 86–90 or neck-motif). The intensity of the GFP signal at laser-cut sites was normalized to that outside of the laser-cut sites for the relative enrichment. The mean values and standard deviations are shown (N = 5, **p<0.01).

The online version of this article includes the following figure supplement(s) for figure 2:

**Figure supplement 1.** The co-localization and co-migration of Kif2C and 53BP1 foci.

**Figure supplement 2.** Kif2C recruitment to DNA damage sites.

**Figure supplement 3.** Kif2C associations with repair proteins were disrupted by mutations.

In line with the involvement of Kif2C in the DDR, Kif2C depletion significantly enhanced the response of HeLa cells to DNA damage treatment, as judged by both reduced cell viability and increased cell death (*Figure 3I and J*). A similar effect was confirmed also in SCC38 cells (*Figure 3—figure supplement 3B and C*), or in HeLa cells with DHTP treatment (*Figure 3—figure supplement 3D*). WT, but not Δ86–90, Kif2C rescued etoposide sensitivity in Kif2C knockout cells (*Figure 3K*), confirming the direct involvement of Kif2C in the DDR.

## Kif2C is required for efficient DSB repair via both HR and NHEJ

To assess further the impact of Kif2C on DNA repair, the kinetics of γ-H2AX post-IR treatment was probed in control and Kif2C depleted cells. Compared to the control HeLa cells, those treated with Kif2C siRNA exhibited more sustained γ-H2AX (*Figure 4A–C*). A similar effect was observed when comparing Kif2C knockout U2OS cells to control U2OS cells (*Figure 4D–F*). The DNA repair deficiency caused by Kif2C depletion was also confirmed using single cell electrophoresis (*Figure 4—figure supplement 1*). Next, we sought to evaluate the impact of Kif2C on specific DSB repair pathways. The repair activity of NHEJ and HR was measured using an intra-chromosomal, I-SceI-induced NHEJ assay and an intra-chromosomal I-SceI-induced HR reporter system, respectively (*Gunn and Stark, 2012*) (*Figure 4G and H*). Interestingly, Kif2C depletion reduced both NHEJ and HR by 3–5 fold (*Figure 4G and H*).

## Kif2C mediates the movement of DSBs, and the formation, fusion, and resolution of DNA damage foci

It is very intriguing how Kif2C promotes DSB repair, given that Kif2C is unlikely to function as a core factor for both NHEJ and HR. We speculated that Kif2C might function in regulation of DSB movement and dynamics in light of several

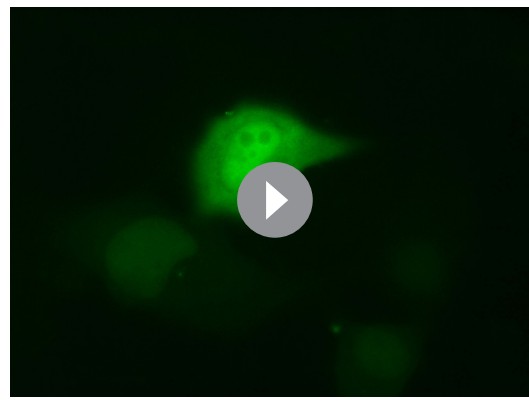

**Video 1.** Kif2C is recruited to DNA damage sites.
https://elifesciences.org/articles/53402#video1

**Video 2.** Kif2C foci co-localize and co-migrate with 53BP1 foci.
https://elifesciences.org/articles/53402#video2

existing findings. First, emerging evidence in both yeast and mammalian cells indicated increased chromatin mobility at sites of DNA DSBs (*Chuang et al., 2006*; *Krawczyk et al., 2012*; *Lemaître and Soutoglou, 2015*; *Levi et al., 2005*; *Lottersberger et al., 2015*; *Marnef and Legube, 2017*), but the underlying mechanism is largely unclear. Second, MTs are well known to support intracellular trafficking of proteins, chromosomes, and other materials, and kinesins are known to produce mechanical work from ATP hydrolysis (*Dogterom et al., 2005*; *Forth and Kapoor, 2017*; *Maizels and Gerlitz, 2015*). Recent studies showed that MT dynamics enhanced the motion of chromatin, especially telomeres, in response to DNA damage (*Lawrimore et al., 2017*; *Lottersberger et al., 2015*). Third, we showed that the MT depolymerase activity of Kif2C, mediated by ATP hydrolysis and tubulin-binding, is required for the prevention of γ-H2AX accumulation. To investigate this potential role of Kif2C, we quantified the mobility of etoposide-induced DSBs, as marked by GFP-53BP1 foci. The 3D trajectories of unbiasedly selected foci were tracked to determine the distance traveled by these foci (*Figure 5A*). We observed that Kif2C depletion, or inhibition of its MT depolymerase activity by DHTP, impaired the mobility of DSBs (*Figure 5B–D*). This effect of Kif2C suppression was comparable to that of Taxol treatment which inhibits MT dynamics and was previously shown to retard DSB movement (*Figure 5C and D*) (*Lottersberger et al., 2015*). To clarify if Kif2C specifically regulates the mobility of damaged chromatin, we analyzed the movement of Centromere Protein B (CENP-B) and Pre-MRNA Processing Factor 6 (PRPF6), as controls. Both CENP-B and PRPF6 form distinct punctate foci in the nucleus that are not DNA damage-induced, and their motilities can indicate undamaged, general chromatin dynamics, and general intra-nuclear dynamics, respectively. Intra-nuclear CENP-B and PRPF6 foci are relatively less dynamic than 53BP1 foci, and there was no significant difference between WT and Kif2C KO or DHTP treatment (*Figure 5—figure supplement 2A and B*). On the other hand, Taxol treatment reduced foci dynamics of CENP-B and PRPF6 (*Figure 5—figure supplement 3*), suggesting a general effect in nuclear dynamics. These data demonstrated that Kif2C mediates DNA damage mobility in a specific manner without affecting other cellular dynamics in general.

Formation of DNA damage foci is a landmark feature of the DDR, but the precise mechanism of this process is still unknown (*Huen and Chen, 2010*). Previous studies analyzing these foci as potential repair centers suggested the clustering of multiple DSB ends and the subsequent formation of macro-domains (*Asaithamby and Chen, 2011*; *Aten et al., 2004*; *Aymard et al., 2017*; *Neumaier et al., 2012*; *Roukos et al., 2013*). In yeast cells, persistent DSBs roam within the nucleus to form these repair centers (*Lisby et al., 2003*; *Marnef and Legube, 2017*). Mammalian DSBs were shown to travel for a similar distance (~2 µM) as yeast DSBs. However, due to a much larger volume of the mammalian nucleus, mammalian DSBs do not roam within the nucleus, but join each other in close proximity (*Marnef and Legube, 2017*; *Neumaier et al., 2012*; *Roukos et al., 2013*). We analyzed the dynamics of DNA damage foci using high-resolution, live-cell imaging (*Figure 5E*). Interestingly, Kif2C depletion or pre-treatment with DHTP or Taxol reduced the formation of DNA damage foci (*Figure 5F*), despite the level of DNA damage being rather elevated under Kif2C suppression (*Figure 5—figure supplement 1B*). To further assess the impact of Kif2C on the dynamics of DNA damage foci, we first allowed the establishment of DNA damage foci (*Figure 5G*), and then challenged cells with DHTP or Taxol. Interestingly, we observed that the occurrence of foci fusion events was decreased by Kif2C depletion or inhibition (*Figure 5H*, *Video 3*); furthermore, foci resolution (disappearance) was also markedly suppressed by these treatments (*Figure 5I and J* and *Video 4*). Presumably, foci fusion represents the movement of DSBs to form larger repair centers/foci, and foci disappearance reflects DNA repair or reassembly of foci. Together, we showed that Kif2C mediates the formation and dynamics of DNA damage foci.

## PARP1 and ATM regulate DSB dynamics largely via Kif2C

As we showed that both PARP1 and ATM act upstream to mediate the recruitment of Kif2C to DNA damage sites, we speculated that PARP1 and ATM play a role in regulation of DSB movement and DNA damage foci formation. Indeed, inhibition of PARP and ATM both significantly reduced the mobility of GFP-53BP1 foci (*Figure 6A and C*). Interestingly, PARP or ATM inhibition only

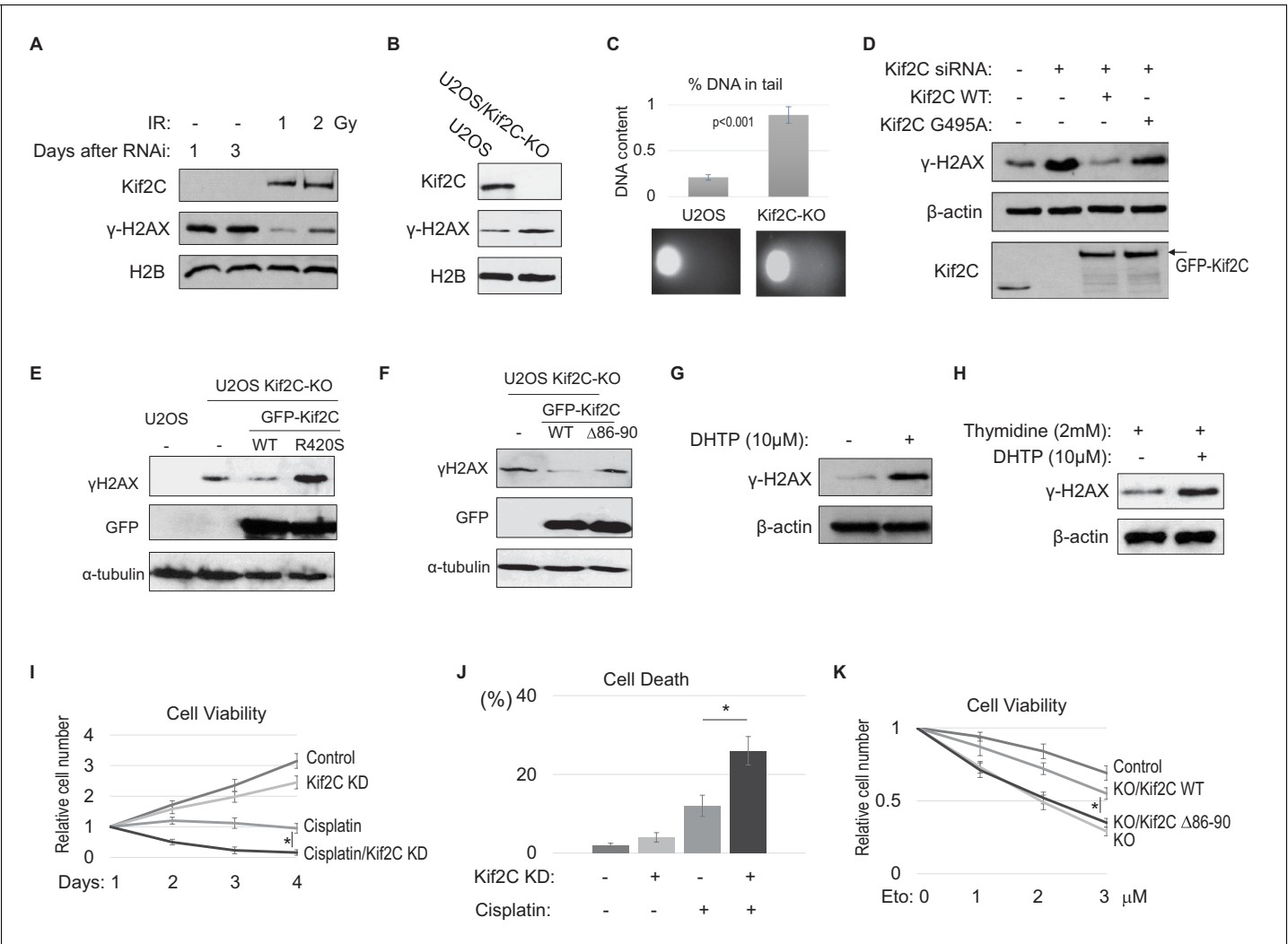

**Figure 3.** Kif2C suppression leads to accumulation of endogenous DNA damage and DNA damage hypersensitivity. (**A**) HeLa cells were treated with Kif2C siRNA for 1 and 3 days, or with IR at 1 or 2 Gy (followed by 30 min incubation), as indicated. These cells were then harvested and analyzed by immunoblotting. (**B**) Kif2C gene deletion was carried out using the CRISPR-Cas9 technique in U2OS cells. Cell lysates were collected and analyzed by immunoblotting. (**C**) The comet assay was performed in control or Kif2C knockout (KO) U2OS cells, as described in Materials and methods. The percentage of DNA in the tail section was quantified, the mean values and standard derivations are shown (N > 20). Representative images are shown below. (**D**) HeLa cells were treated with control siRNA or Kif2C siRNA, and reconstituted with siRNA resistant GFP-Kif2C (WT or G495A), as indicated. Cell lysates were harvested and analyzed by immunoblotting. (**E**) Control or Kif2C knockout (KO) U2OS cells were transfected with WT or R420S Kif2C tagged with GFP, as indicated. One day after transfection, the samples were analyzed by immunoblotting. (**F**) U2OS Kif2C knockout (KO) cells were transfected with WT or Δ86–90 Kif2C tagged with GFP, as indicated. One day after transfection, the samples were analyzed by immunoblotting. (**G**) Asynchronized HeLa cells were treated with 20 μM DHTP for 3 hr, as indicated. The cell lysates were analyzed by immunoblotting. (**H**) HeLa cells were first synchronized at G1/S by thymidine-arrest, and then treated with 10 μM DHTP for 3 hr. The cell lysates were analyzed by immunoblotting. (**I**) HeLa cells were incubated in cisplatin (6.7 μM) and Kif2C siRNA, as indicated. The relative cell viability was determined by normalizing the cell number to that of the first day. The mean values and standard deviations, calculated from three independent experiments, are shown. *p<0.05. (**J**) HeLa cells were treated as in panel I for 2 days, and measured by the trypan blue exclusion assay for cell death. The mean values and standard deviations, calculated from three independent experiments, are shown. (**K**) WT or Δ86–90 Kif2C was expressed in Kif2C KO cells as in panel F. 1 day after transfection, these cells, along with control U2OS, were treated with various doses of etoposide for 2 days. The relative cell viability was determined by first calculating the ratio of cell number in day 3 to that in day 1, and then normalizing the ratio of etoposide treated cells to that of the untreated. The mean values and standard deviations, calculated from three independent experiments, are shown.

The online version of this article includes the following figure supplement(s) for figure 3:

**Figure supplement 1.** Foci formation of γ-H2AX and 53BP1 in undamaged Kif2C knockout (KO) cells.
**Figure supplement 2.** Kif2C recruitment and cell cycle effect.
**Figure supplement 3.** The MT depolymerase activity of Kif2C is involved in the DNA damage response.

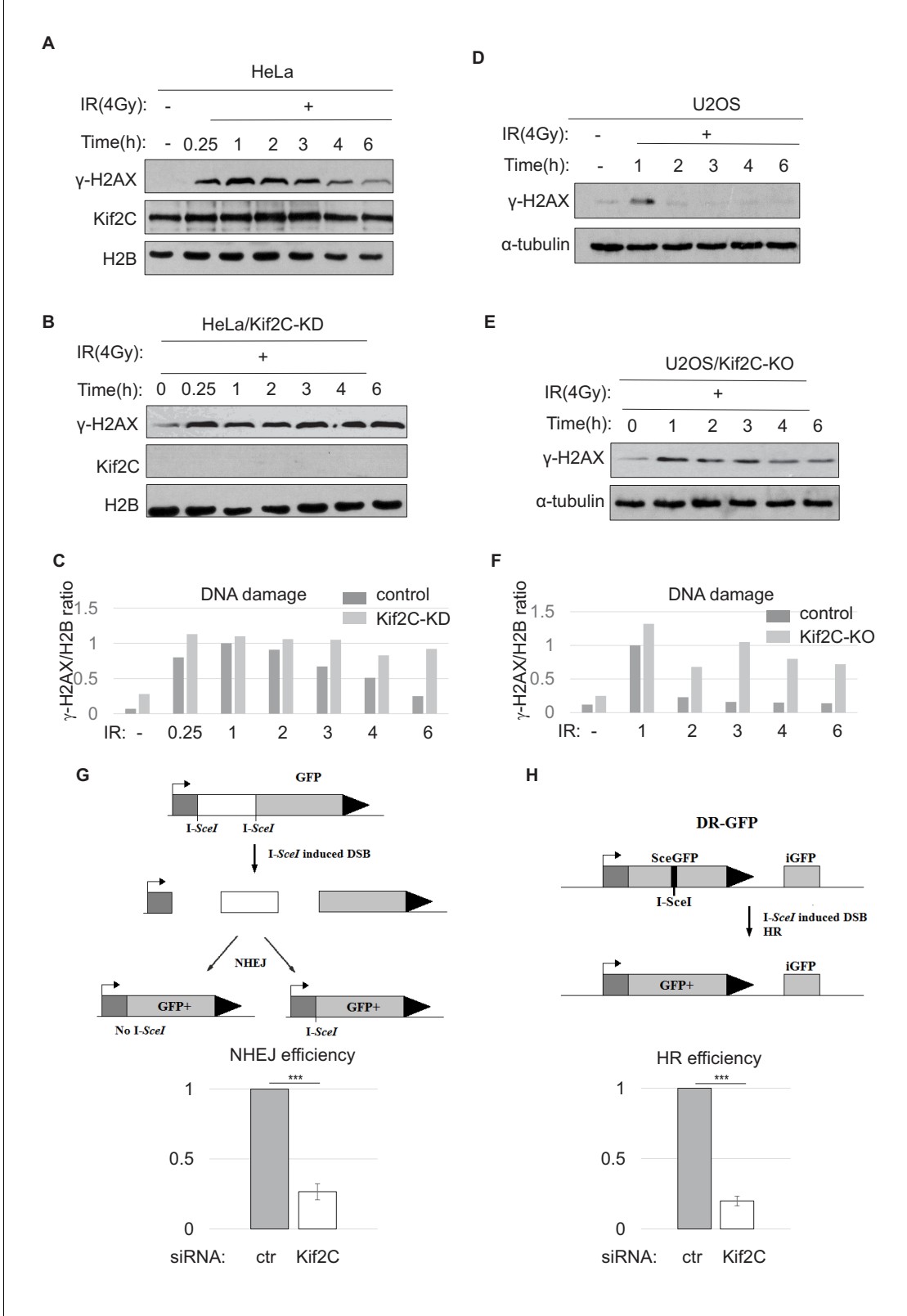

**Figure 4.** Kif2C is required for DNA double strand break repair. (**A, B, C**) HeLa cells treated with control (**A**) or Kif2C siRNA (**B**) were irradiated with 4 Gy IR, and incubated as indicated. The cell lysates were analyzed by immunoblotting. Quantification is shown in panel C. (**D, E, F**) U2OS cells, control (**D**) or Kif2C knockout (KO, (**E**), were irradiated with 4 Gy IR, and incubated as indicated. The cell lysates were analyzed by immunoblotting. Quantification is shown in panel F (as relative to 1 hr time point in control HeLa cells). (**G, H**) Chromosome-integrated, I-SceI-induced NHEJ (**G**) or HR (**H**) reporter

*Figure 4 continued on next page*

*Figure 4 continued*

systems are illustrated in the upper panels. These reporter cells were transfected with control or Kif2C siRNA. DNA repair was measured by immunoblotting of GFP expression in relative to β-actin expression. The GFP/β-actin ratio in Kif2C-depleted cells was normalized to that in control cells for relative repair efficiency. The mean values and standard deviations, calculated from three independent experiments, are shown. Statistical significance was analyzed using an unpaired 2-tailed Student's t-test (***$p<0.001$). Kif2C depletion did not impact the expression of I-SceI (*Figure 4—figure supplement 2*).

The online version of this article includes the following figure supplement(s) for figure 4:

**Figure supplement 1.** Kif2C depletion impairs DNA repair.

**Figure supplement 2.** Kif2C depletion did not affect the expression of I-SceI.

moderately retarded GFP-53BP1 foci movement in Kif2C depleted cells (*Figure 6C*), indicating that PARP and ATM govern DSB mobility largely, although not exclusively, through Kif2C. On the other hand, these data also indicated that Kif2C depletion did not further suppress the mobility of GFP-53BP1 in cells with PARP or ATM inhibition (*Figure 6—figure supplement 1*), suggesting that the function of Kif2C in this process is dependent on both PARP1 and ATM, which act upstream to mediate the DNA damage recruitment of Kif2C.

## Discussion

### Kif2C is a new player of the DNA damage response

As a member of the MT depolymerase family, Kif2C was shown to govern several aspects of cell division in mitosis, including spindle assembly, chromosome congression, and kinetochore-MT attachment (*Manning et al., 2007*; *Sanhaji et al., 2011*). The role of Kif2C in interphase cells is less characterized, despite that its dominant localization in the nucleus suggests possible functions of Kif2C in intra-nuclear processes. Interestingly, we reported here a direct involvement of Kif2C in DNA repair, as a previously undefined interphase function of Kif2C. First, we showed that Kif2C associated with DSB-mimicking structures in *Xenopus* egg extracts and human cell lysates. Consistently, Kif2C bound several established DNA repair factors, including PARP1, H2AX, and Ku70/80. Second, Kif2C was recruited to DNA damage sites in interphase cells via two distinct mechanisms. The initial recruitment of Kif2C occurred within seconds in a PAR-dependent manner, whereas the sustained localization of Kif2C at DNA damage sites was disrupted by ATM inhibition. Thus, we characterized Kif2C as a downstream factor of the PARP1 and ATM-mediated DNA damage responses. Third, Kif2C was required for efficient DNA DSB repair via both NHEJ and HR; consequently, depletion or inhibition of Kif2C leads to both accumulation of endogenous DSB and DNA damage hypersensitivity. Interestingly, a Kif2C mutant (Δ86–90) specifically deficient in DNA damage recruitment was unable to rescue DSB accumulation or etoposide-sensitivity in Kif2C depleted cells. Furthermore, Kif2C inhibition led to DSB accumulation in cells synchronized at G1/S. Together, our studies revealed a new role of Kif2C in facilitating DNA DSB repair that is distinct from its known functions in mitotic progression.

### Kif2C mediates the mobility of DSBs, and the formation of DNA damage foci

While the core pathways of NHEJ and HR have been well studied, an emerging topic of DNA repair lies in the spatiotemporal dynamics of DSBs (*Hauer and Gasser, 2017*; *Lemaître and Soutoglou, 2015*; *Marnef and Legube, 2017*; *Miné-Hattab and Rothstein, 2013*). In particular, clustering of DSB ends into 'repair centers' has been observed for longer than a decade; and the increased mobility of DSB ends within the nucleus has been reported in yeast and mammalian cells (*Chuang et al., 2006*; *Chung et al., 2015*; *Krawczyk et al., 2012*; *Lemaître and Soutoglou, 2015*; *Levi et al., 2005*; *Lottersberger et al., 2015*; *Marnef and Legube, 2017*; *Neumaier et al., 2012*). However, mechanistic understandings of these phenomena are largely absent within the context of current DDR regulators. We revealed in this study that a specific kinesin motor protein, Kif2C, directly promotes DSB mobility and mediates the formation and fusion of DNA damage foci. Our findings suggested a model that, upon recruitment to DSBs, Kif2C propels the physical movement of damaged chromatin to promote DNA repair, in a manner that relies on the ATPase and tubulin-binding

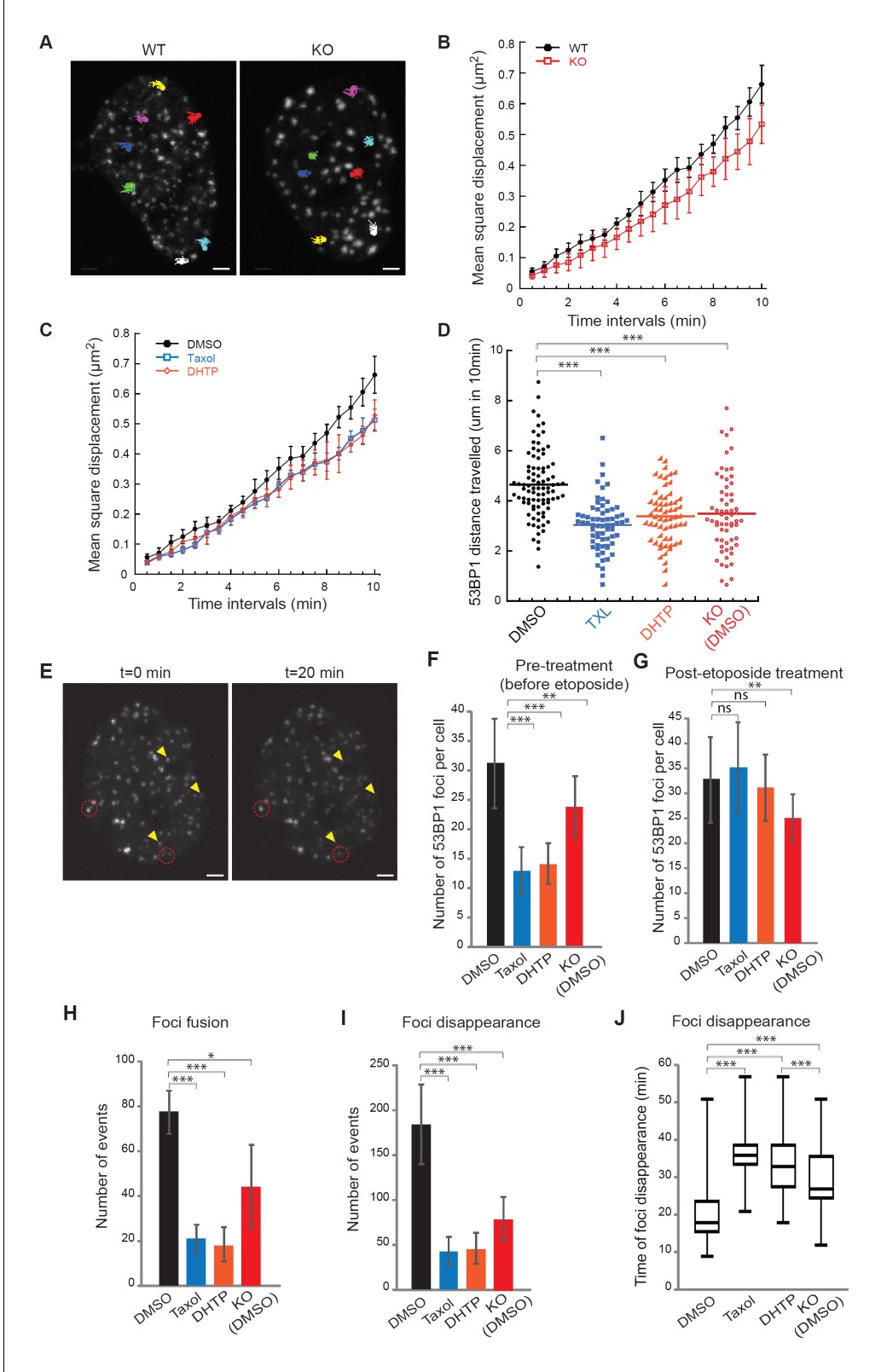

**Figure 5.** Kif2C mediates DNA double strand break mobility and foci dynamics. (**A**) Examples of 10 min mobility traces of EGFP-53BP1 foci in WT and Kif2c knockout (KO) U2OS cells after etoposide (20 µM) treatment. Kif2C depletion did not impact 53BP1 expression (***Figure 5—figure supplement 1A***). (**B**) Mean-square displacement measurements of EGFP-53BP1 foci in WT and Kif2C KO U2OS cells, shown in black in red, respectively. (**C**) Mean-square displacement measurements of EGFP-53BP1 foci in WT U2OS cells treated with the vehicle control (DMSO), Taxol (5 µM), or DHTP (20 µM), as

*Figure 5 continued on next page*

*Figure 5 continued*

indicated. (D) Quantification of the distance travelled by EGFP-53BP1 foci over 10 min in the corresponding cells described in B-C. (E) Examples of disappearance (yellow arrowheads) and fusion (red circle) events of EGFP-53BP1 foci induced by etoposide in U2OS cells. (F, G) Number of EGFP-53BP1 foci in WT or Kif2C KO U2OS cells, treated with the vehicle control (DMSO), Taxol, or DHTP. These inhibitors were added either 5 min before (F) or 5 min after (G) etoposide treatment. (H–J) Numbers of fusion (H) and disappearance (I–J) events of EGFP-53BP1 foci in the corresponding cells in panel G are shown. A total of 15 randomly selected cells were analyzed over three independent experimental runs. For disappearance events, number of occurrence in the first 30 min under each treatment condition is shown in (I) and the time required for foci disappearance (min) over the entire hour of recording is shown in (J) (>150 events quantified per condition). The box represents 50% of the foci disappearance events and the line shows the median of the data set. All microscopy image acquisitions began five minutes after final compound treatment, either every 30 s for 10 min (A–D) or every 3 min for one hour (H–J). All data were collected from at least three independent experimental sets. Error bars, S.D.; ns: p>0.05; *p≤0.05; **p≤0.01; ***p≤0.001, by Student's *t*-test.

The online version of this article includes the following figure supplement(s) for figure 5:

**Figure supplement 1.** Kif2C depletion did not affect the expression of GFP-53BP1 (A).
**Figure supplement 2.** Kif2C depletion did not influence the general nuclear dynamics.
**Figure supplement 3.** Taxol, but not DHTP, reduced the general mobility of CENP-B and PRPF6.

activities of Kif2C; Kif2C facilitates the formation of DNA damage foci, which potentially involves the mobility and clustering of DSBs, as shown previously (*Asaithamby and Chen, 2011*; *Aten et al., 2004*; *Aymard et al., 2017*; *Neumaier et al., 2012*; *Roukos et al., 2013*). In addition to foci formation, we observed also the occurrence of DSB foci fusion and resolution, indicating that DSBs may undergo dynamic organization and reorganization during DNA repair. These events are reduced by Kif2C depletion or inhibition, thus implicating a role of Kif2C in these processes.

In addition to the underlying mechanism, the functional impact of DSB mobility and foci formation remains to be better clarified. It has been generally hypothesized that this pattern of DSB dynamics facilitates DSB repair, for example by keeping DSB ends in close proximity, and increasing the local concentration of repair proteins (*Lottersberger et al., 2015*; *Miné-Hattab and Rothstein, 2012*; *Miné-Hattab and Rothstein, 2013*). Furthermore, mounting evidence suggested that DSB mobility may enable homology search during HR (*Marnef and Legube, 2017*; *Miné-Hattab and Rothstein, 2012*; *Schrank et al., 2018*). On the other hand, MT and the linker of the nucleoskeleton and cytoskeleton (LINC)-mediated DSB mobility was shown to promote NHEJ of dysfunctional telomeres (*Aymard et al., 2017*; *Lottersberger et al., 2015*). By characterizing Kif2C as a specific regulator of DSB mobility, our study provided an opportunity to assess the functional involvement of DSB dynamics in repair. Interestingly, we demonstrated that Kif2C is required for the efficient DSB repair via both HR and NHEJ. Future studies shall be directed to determine more precisely how Kif2C mediates DSB dynamics, and how this process may interact with the core repair machinery of HR and NHEJ.

We showed that the initial or sustained recruitment of Kif2C to DNA damage sites is dependent on PARP1 or ATM activation, respectively. Thus, we set out to investigate if either PARP1 or ATM governs DNA damage dynamics via Kif2C. Of note, ATM was shown to govern DSB mobility in previous studies (*Becker et al., 2014*; *Dimitrova et al., 2008*). PARP1 is known to play a crucial role in sensing DNA damage, recruiting repair factors, and modulating chromatin structure, but its involvement in DSB movement was not reported. We clarified in our study that PARP1 and ATM inhibition markedly retarded DSB mobility. Inhibition of PARP1 or ATM in Kif2C-depleted cells less significantly affected DSB mobility, validating that Kif2C is a major downstream of PARP1 and ATM in regulation of DSB mobility, but at the same time, suggesting the existence of redundant pathways.

## The emerging role of microtubule dynamics in DNA repair

Since the first report of nuclear actin in *Xenopus*, the existence of the actin network in the nucleus, and its function in nuclear architecture and genomic regulation, have been well recognized (*Belin et al., 2015*; *Caridi et al., 2018*; *Grosse and Vartiainen, 2013*; *Misu et al., 2017*; *Schrank et al., 2018*). By comparison, MT assembly is viewed as a cytoplasmic event, except in mitosis after nuclear envelop breakdown. Thus, the function of Kif2C, a MT

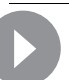

**Video 3.** An example of foci fusion.
https://elifesciences.org/articles/53402#video3

**Video 4.** An example of foci resolution.
https://elifesciences.org/articles/53402#video4

depolymerase, in DNA repair is largely unexpected. In particular, our studies using established Kif2C mutants and inhibitor suggested that the ATPase and tubulin-binding activities of Kif2C were indispensable for suppressing DNA damage accumulation. Potentially in line with our findings, previous studies showed that MT poisons caused endogenous DNA damage and reduced DNA repair (*Branham et al., 2004*; *Lottersberger et al., 2015*; *Poruchynsky et al., 2015*; *Rogalska and Marczak, 2015*).

While cytoplasmic MTs can indirectly influence the DDR, for example, via the nuclear import of repair factors (*Poruchynsky et al., 2015*), or via the LINC complex (*Aymard et al., 2017*; *Lottersberger et al., 2015*), our study suggested a rather direct involvement of nuclear MT components in the DDR. This is particularly relevant as many kinesins, as well as low levels of tubulins, are present in the nucleus (*Akoumianaki et al., 2009*; *Kırlı et al., 2015*; *Kumeta et al., 2013*). Interestingly, Kif4A and γ-tubulin were shown to associate with Rad51 and possibly other repair proteins (*Lesca et al., 2005*; *Wu et al., 2008*). Yeast kinesin-14 and nuclear pore proteins mediate the perinuclear tethering of telomeric DSBs in yeast cells (*Chung et al., 2015*). Moreover, recent evidence demonstrated that inhibitors of MT assembly reduced the mobility of DSBs (*Lawrimore et al., 2017*; *Lottersberger et al., 2015*), further suggesting a nuclear function of MTs.

To account for the potential role of MT dynamics in DNA repair, a provocative possibility is that MT assembly occurs in the nucleus after DNA damage, presumably at a low and transient level. Along this line, a previous study visualized increased tubulin nucleation and MT rearrangement after DNA damage, although it was not defined if this event occurs at least partially in the nucleus (*Porter and Lee, 2001*). A study in yeast cells detected the assembly of long and stable MTs in the interphase nuclei when cell enters quiescence (*Laporte et al., 2013*); a more recent study visualized DNA damage-inducible intra-nuclear microtubule filaments (DIMs) in yeast cells using GFP-tagged tubulin (*Oshidari et al., 2018*). However, the formation of detectable DIMs in mammalian cells remains to be demonstrated. On the other hand, as an alternative hypothesis to be considered, MT filament assembly via tubulin nucleation may not occur in the nucleus of damaged mammalian cells, but rather, certain regulators and mechanisms of MT assembly/disassembly are employed by the DDR machinery to govern the dynamic movement and repair of broken DNA ends. In all cases, the characterization of Kif2C as a new DDR factor that mediates DNA damage movement and foci formation sheds new light on the spatiotemporal regulation of DNA damage dynamics. Future studies building on these findings shall further delineate the involvement of MT regulators in the DNA damage response.

## Materials and methods

### Cell culture, transfection and treatment

Human cervix carcinoma (HeLa) and bone osteosarcoma epithelial (U2OS) lines, authenticated by ATCC, were maintained in Dulbecco's modified Eagle medium (DMEM, Hyclone) with 10% fetal bovine serum (FBS, Hyclone). Human head and neck squamous cell carcinoma UM-SCC-38 cells were authenticated and maintained as in previous studies (*Brenner et al., 2010*; *Wang et al., 2012a*). Cell viability and death assays were performed as in our previous study (*Wang et al., 2014*). Briefly, cells were incubated for 1–4 days. The numbers of viable cells were counted using a hemocytometer. To measure cell death, trypan blue staining was performed by mixing 0.4% trypan blue in PBS with cell suspension at a 1:10 ratio. Ionized radiation was performed using an X-ray cabinet (RS-2000 Biological irradiator). Transfection of expression vectors was carried out using Lipofectamine 2000 (Invitrogen) or TransIT transfection reagents (from Mirus Bio) following the protocol recommended by the manufacturer. SiRNA targeting Kif2C (5-AUCUGGAGAACCAA GCAU-3', Integrated DNA Technologies) was transfected into cells using Lipofectamine RNAi MAX (Invitrogen). A non-targeting control siRNA was used as a control. Kif2C knockout U2OS cells was generating using the established CRISPR-cas9 gene editing method with following single guide (sg) sequence: GCAAGC TGACACAGGTGCTG in lentiCRISPR v2 vector (a gift from Feng Zhang via Addgene plasmid # 52961) (*Sanjana et al., 2014*). Knockout efficiency was assessed by TIDE (Tracking of Indels by

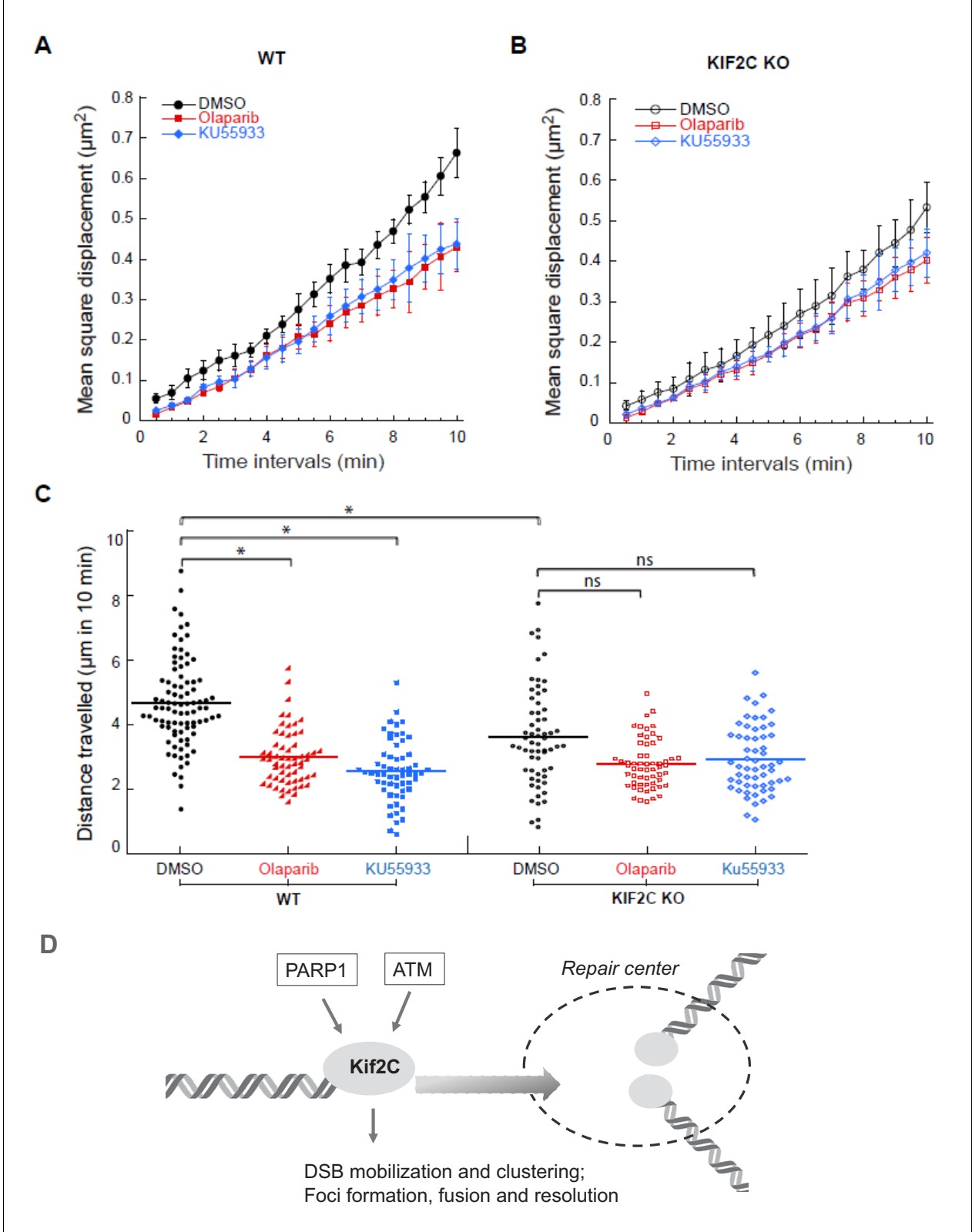

**Figure 6.** ATM and PARP inhibition impairs Kif2C-dependent foci mobility. (A–B) Mean-square displacement measurements of EGFP-53BP1 foci in WT (A) and Kif2C KO (B) U2OS cells treated with the vehicle control (DMSO), the PARP1 inhibitor olaparib (10 µM), or the ATM inhibitor KU55933 (20 µM), as indicated. More than 50 foci were analyzed in three independent experiments. (C) Quantification of the distance travelled by EGFP-53BP1 foci over 10 min in the corresponding cells described in A-B. Experimental set up and image acquisition were the same as described in *Figure 5*. (D) Kif2C

*Figure 6 continued on next page*

*Figure 6 continued*

mediates DSB end mobilization and the formation of DNA damage foci (model). Kif2C is recruited to DSB ends in a manner that depends on both ATM and PARP activities. Mediated by its MT depolymerase activity, Kif2C promotes the movement, and the subsequent clustering, of DSB ends. Therefore, Kif2C is an important downstream factor of the PARP and ATM-mediated DNA damage response that governs the mobility and dynamics of DSB ends.

The online version of this article includes the following figure supplement(s) for figure 6:

**Figure supplement 1.** ATM and PARP inhibition impairs Kif2C-dependent foci mobility.

Decomposition) analysis using the web tool (https://tide.deskgen.com/) and confirmed by western blot.

## Cloning and mutagenesis

*Xenopus* Kif2C gene was cloned from a *Xenopus* oocyte cDNA library, and inserted into a pMBP vector with an N-terminal MBP-tag. Kif2C G495A, R420S, Δ86–90, Δneck, and siRNA-resistant mutants were generated using site-directed mutagenesis (Agilent) following the protocol recommended by the manufacturer. The human Kif2C expression vector was obtained from Addgene (mEmerald-MCAK-C-7, a gift from Michael Davidson via Addgene, plasmid # 54161).

## DNA binding assay

Biotin-labeled double strand DNA fragment (dsDNA, 500 bp) was generated using biotin-11-ddUTP (Thermo Scientific, #R0081) incorporation, and PCR amplification using Taq polymerase and a pMBP vector (as template). Biotin-labeled DNA (produced as above) or biotin dA-dT (70 mer) was conjugated on streptavidin magnetic beads (New England Biolabs, #S1420S) and incubated in *Xenopus* egg extracts and HeLa cell lysates. The beads were re-isolated using a magnet, washed five times, and then resolved by SDS-PAGE.

## HR and NHEJ assays

Homologous recombination assay was performed in a HeLa-derived cell line stably integrated with a DR-GFP reporter cassette (a gift from Dr. Jeffrey Parvin at the Ohio State University). The reporter consisted of direct repeats of two differentially mutated green fluorescent proteins (GFP), Sce GFP and iGFP. SceGFP contains an I-SceI recognition site and in-frame termination codons. An 812 bp internal GFP fragment (iGFP) was used by HR to repair the DSB. Briefly, cells were seeded at $3 \times 10^5$ cells per well in a 6-well plate one day before siRNA treatment. After removing the siRNA, the cells were grown for 48 hr in fresh medium and transfected with an expression vector of I-SceI endonuclease (a gift from Dr. Maria Jasin at Memorial Sloan Kettering Cancer Center). In this assay, a full-length GFP is expressed only after DSBs introduced by I-SceI endonuclease are repaired by HR, and the level of full-length GFP (and control β-actin) expression was quantified by immunoblotting and NIH ImageJ.

The NHEJ assay was performed in U2OS-EJ5 cells (a gift from Dr. Jeremy Stark at the Beckman Research Institute of the City of Hope). Briefly, cells were seeded at $3 \times 10^5$ cells per well in a 6-well plate 24 hr before siRNA treatment. After removing the siRNA, the cells were grown for 48 hr in fresh medium and transfected with an expression vector of I-SceI endonuclease. In this assay, GFP is expressed only after DSBs introduced by I-SceI endonuclease are repaired by NHEJ, and the level of GFP expression was quantified by immunoblotting and NIH ImageJ. In both HR and NHEJ assays, approximately 3–10% cells in the control-treated groups exhibited GFP-positive.

## Immunoblotting

Sodium dodecyl sulfate-polyacrylamide gel electrophoresis (SDS-PAGE) and immunoblotting were carried out as previously described (*Ren et al., 2017*), using the following antibodies: γ-H2AX (A300-081A-M), and Ku80 (A302-627A-T) from Bethyl Laboratories (Montgomery, TX); ATM (sc-377293), DNA-PKcs (sc-390849), GFP (sc-9996), γ-H2AX (sc-517348), Kif2C (sc-81305), and Ku70 (sc-56129), from Santa Cruz Biotechnology (Dallas, TX); H2B (ab1790-100) and α-tubulin (ab7291) from Abcam (Cambridge, MA); β-actin (#4970T) from Cell Signaling Technology (Beverly, MA); and Artemis (GTX100128) from Genetex (Irvine, CA).

## Immunofluorescence and imaging

Cells were grown on coverglasses, washed with PBS twice, and fixed with 3% formaldehyde with 0.1% Triton X-100 for 30 min. 0.05% Saponin containing PBS was used to permeabilize the fixed cells followed by blocking with 5% goat serum for 30 min. Primary antibodies were diluted in blocking buffer and incubated with the cells for 2 hr. The cells were then incubated with Alexa Fluor secondary antibodies (Invitrogen, 1: 2,000) for 1 hr at room temperature. The nuclei of cells were stained with 4',6-diamidino-2-phenylindole (DAPI), and the stained cells were imaged using a Zeiss Axiovert 200M inverted fluorescence microscope at the UNMC Advanced Microscopy Core Facility. Laser micro-irradiation was performed using 405 nm laser under the Zeiss Axiovert 200M Microscope with Marianas Software (Intelligent Imaging Innovations, Inc Denver, CO).

## Microscopic analysis of DNA damage foci mobility and dynamics

EGFP-53BP1 (or mApple-53BP1, or Control foci constructs: EGFP-PRPF6 or Cenp-B-mCherry)-transfected cells were seeded in ibidi μ-Dish 35 mm Quad dish the day prior to imaging. Formation of 53BP1 foci was induced by the addition of 20 μM etoposide. Other compounds such as taxol or DHTP were added to the cells either prior to (pre-treatment) or after (post-treatment) etoposide treatment. Image acquisition was carried out using a Zeiss spinning disk confocal microscopy system equipped with a 63 × *PlanAprochromat oil* objective. After cells expressing those constructs were located and the imaging positions were selected, microscopy recordings were then started (usually 5 min after the last treatment, for consistency reason). Imaging of the control foci, that is CENP-B-mCherry and EGFP-PRPF6, was done in a similar manner except that their formation does not require etoposide addition. For foci mobility, time-lapse recordings were done every 30 s for 10 min. For foci disappearance or fusion, recordings were done every 3 min for one hours. Z-stack images were acquired at 0.5 μm intervals covering a range from 6 to 8 μm. Foci tracking was done using all the acquired stacks for positional information using ImageJ (NIH), and the foci number was quantified using the automatic particle counting option. For image presentation in figure panels, 2D-maximum intensity projection images were generated using the ZEN blue software. Data analysis and graph presentations were performed using Excel (Microsoft) and KaleidaGraph (Synergy). Student's t-test was used for statistical analysis. Mean-square displacement was calculated as previously described (*Lottersberger et al., 2015*) using the following equation

$$MSD(\text{t}) = \frac{1}{n}\sum_{i=1}^{n} Di(\text{t})^2$$

Where

$$Di(t) = \sqrt{\left(\left(x_t^i - x_t^{GC}\right) - \left(x_{t-t}^i - x_{t-t}^{GC}\right)\right)^2 + \left(\left(y_t^i - y_t^{GC}\right) - \left(y_{t-t}^i - y_{t-t}^{GC}\right)\right)^2}$$

## Pull-down assay

For protein association studies, MBP Kif2C WT and mutants were expressed in BL21 bacteria cell, purified on amylose beads, and then incubated in HeLa cell lysates for 1 hr at room temperature. The beads were re-isolated using low speed centrifugation, washed five times, and then resolved by SDS-PAGE. For the plasmid DNA pull-down assay in *Figure 1C*, pMBP plasmid was either uncut or linearized by EcoRV endonuclease (New England Biolabs, #R3195). MBP-Kif2C was conjugated on amylose beads and incubated in *Xenopus* egg extracts supplemented with either uncut or linearized pMBP plasmid for 1 hr at room temperature. The beads were re-isolated using low speed centrifugation, washed five times, and then boiled in distilled water. The samples were used as templates for PCR with Taq Polymerase.

## Single cell gel electrophoresis (comet assay)

Cells were washed with PBS, trypsinized, and plated in 0.65% low melting agarose. After solidification, slides were incubated in lysis solution (1 M NaCl, 3.5 mM N-laurylsarcosine, 50 mM NaOH) for 2 hr. Slides were then washed, and incubated in alkaline electrophoresis buffer (50 mM NaOH, 2 mM EDTA) for 30 min. After electrophoresis for 10 min at 20 V, slides were stained with propidum iodide (25 μg/mL).

### *Xenopus* egg extracts

Eggs were rinsed in distilled water and de-jellied with 2% cysteine in 1x XB (1 M KCl, 10 mM MgCl$_2$, 100 mM HEPES pH 7.7, and 500 mM sucrose). Eggs were washed in 0.2x MMR buffer (100 mM NaCl, 2 mM KCl, 1 mM MgCl$_2$, 2 mM CaCl$_2$, 0.1 mM EDTA, 10 mM HEPES), and activated with Ca$^{2+}$ ionophore. Eggs were then washed and crushed by centrifugation at 10,000 g. The cytoplasmic layer was transferred to new tubes, supplemented with an energy mix (7.5 mM creatine phosphate, 1 mM ATP, 1 MgCl$_2$), and then further separated by centrifugation at 10,000 g for 15 min.

## Acknowledgements

We thank Dr. Jay Reddy (University of Nebraska-Lincoln) for technical support with X-ray irradiation, Christian Charbonneau of the Bioimaging facility of the Institute for Research in Immunology and Cancer (IRIC, Université de Montréal) for technical support with high-resolution microscopy, and Dr. Greg Oakley (University of Nebraska Medical Center) for stimulating discussions. The UNMC Advanced Microscopy Core Facility was supported by the Nebraska Research Initiative, the Fred and Pamela Buffett Cancer Center Support Grant (P30CA036727), and an Institutional Development Award (IDeA) from the NIGMS of the NIH (P30GM106397). The IRIC is supported in part by the Canadian Center of Excellence in Commercialization and Research (CECR), the Canada Foundation for Innovation and Fonds de recherche du Québec – Santé (FRQS). This work was partially supported by NIH grant CA172574 to AP, and funding to BHK from the Canadian Institutes of Health Research (CIHR, PJT# 148982 and 152920), Cancer Research Society (CRS), FRQS and the Strategic Project Committee (SPC) of IRIC.

## Additional information

### Funding

| Funder | Grant reference number | Author |
|---|---|---|
| National Institutes of Health | CA172574 | Aimin Peng |
| Canadian Institutes of Health Research | 148982 | Benjamin H Kwok |
| Canadian Institutes of Health Research | 152920 | Benjamin H Kwok |
| Cancer Research Society | | Benjamin H Kwok |

The funders had no role in study design, data collection and interpretation, or the decision to submit the work for publication.

### Author contributions

Songli Zhu, Conceptualization, Data curation, Methodology, Writing - original draft; Mohammadjavad Paydar, Feifei Wang, Yanqiu Li, Ling Wang, Tadayoshi Bessho, Data curation, Methodology; Benoit Barrette, Resources, Data curation; Benjamin H Kwok, Conceptualization, Formal analysis, Funding acquisition, Writing - review and editing, Supervision; Aimin Peng, Conceptualization, Data curation, Formal analysis, Supervision, Writing - original draft, Writing - review and editing

### Author ORCIDs

Mohammadjavad Paydar (iD) http://orcid.org/0000-0002-0569-7017
Tadayoshi Bessho (iD) http://orcid.org/0000-0001-8719-9646
Benjamin H Kwok (iD) https://orcid.org/0000-0002-3668-7049
Aimin Peng (iD) https://orcid.org/0000-0002-2452-1949

### Decision letter and Author response

Decision letter https://doi.org/10.7554/eLife.53402.sa1
Author response https://doi.org/10.7554/eLife.53402.sa2

## Additional files

### Supplementary files
• Transparent reporting form

### Data availability
All data generated or analysed during this study are included in the manuscript and supporting files.

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
