## [Decision Letter]

**Acceptance summary:**

This interesting manuscript describes the new finding that Kif2C, a microtubule depolymerase, is directly involved in DSB repair, contributing to the DNA damage-induced motion of DSBs. The authors identify that Kif2C binds to linear but not circular DNA in *Xenopus* and HeLa cell extracts. They show that Kif2C localizes to laser-induced DNA damage within 1 min. The initial recruitment is dependent on PARP activity and stable association depends on ATM activity. They were able to map two critical sites in the N-terminal region of Kif2C that are both required for association with laser damage. Using siRNA knockdown and a U2OS knockout cell line the authors show that Kif2C is required to suppress accumulation of γ-H2AX, a marker for DNA damage in cycling cells without exogenous DNA damage. They show by back complementation that the MT depolymerizing activity of Kif2C is required for this effect. Finally, the authors report a small but reproducible effect on viability after cis-Pt treatment of Kif2C knockdown or knockout in multiple cell systems. Consistent with the regulation by PARP and ATM, pharmacological inhibition of PARP and ATM result in a defect of 53BP1 focus mobility in wild type cells and to a lesser degree in the Kif2C knockout cells, suggesting that these regulators target Kif2C and other proteins contributing to 53BP1 focus mobility.

**Decision letter after peer review:**

[Editors’ note: the authors submitted for reconsideration following the decision after peer review. What follows is the decision letter after the first round of review.]

Thank you for submitting your work entitled "Kinesin Kif2C in regulation of DNA double strand break dynamics and repair" for consideration by *eLife*. Your article has been reviewed by 3 peer reviewers including Wolf-Dietrich Heyer as the Reviewing Editor and Reviewer #1, and the evaluation has been overseen by a Reviewing Editor and a Senior Editor.

Our decision has been reached after consultation between the reviewers. Based on these discussions and the individual reviews below, we regret to inform you that your work will not be considered further for publication in *eLife*.

Thank you for giving us the opportunity to fully evaluate your manuscript entitled "Kinesin 1 Kif2C in Regulation of DNA Double Strand Break Dynamics and Repair". The observations you report are of great potential interest about a possible role of this MT depolymerase and nuclear MTs in DNA repair. However, the review process revealed a rather long list of concerns and issues that would need to be addressed in a revision. These concerns are so strong that depending on the outcome of controls and additional experiments, the results could compromise your conclusion that nuclear microtubules and Kif2c play a direct nuclear role in genome stability.

It is the policy of *eLife* to decline submissions, where a revision would presumably take longer than 2 months. The reviewers, which includes two specialists with good knowledge of the advanced cytology you conduct and mammalian cells, think that a satisfactory revision would take considerably longer than 8 weeks.

In this spirit, this submission is rejected, leaving you the choice to either submit the manuscript elsewhere, hopefully with changes based on the attached reviews, or coming back to *eLife* with a new submission that addresses the comments made in the reviews with particular attention to the major highlighted below. I would be happy to handle that renewed submission.

The individual reviews are attached to this letter, I want to highlight here the critical points that would need to be addressed in a new submission, if you choose to go that route.

1) Figure 1: Please address comment 1 by reviewer 1.

Figure 1—figure supplement 1: In the minimum, it should be clarified that the doxorubicin treatment was sufficient to elicit a cellular DNA damage response.

2) Figure 2:

Comment 2 from reviewer 1 should not pose a problem and the data for the co-localization quantitation should exist.

The concern of reviewer 2 in comment 2 about the Kif2CΔ86-90 mutant protein must be addressed. Comment 10 of reviewer 2 constitutes an important control.

3) Figure 3:

Comments 3 and 4 of reviewer 2 are pertinent.

4) Figure 4:

It will be critical to address comment 1 of reviewer 1 and comment 5 of reviewer 2. The taxol and nocadzole experiments (comment 6 of reviewer 2) would be nice additions but are not essential.

5) Figure 5, Figure 6: There are significant concerns by reviewers 2 and 3 about the MSD analysis that must be addressed. In particular, it will be critical to distinguish between a nuclear effect and an overall cellular effect. All reviewers consider this to be the most critical and time-consuming part of the potential revision.

6) I want to stress that a new submission does not need to address the issue of evidence for nuclear microtubules during repair. We believe that would be a logical next step and possibly achievable in Kif2C knockout or knockdown cell, but beyond the scope of the current manuscript.

7) There are many minor comments and corrections, including comments on literature citations that should be addressed but require only text changes or additions.

8) Please note especially the comments about missing statistical validation in Figure 2, Figure 5 and Figure 6, which will be critical to address.

We hope that this list along with the detailed reviews provides sufficient guidance for a possible revision.

Reviewer #1:

This is an interesting manuscript describing the novel finding that Kif2C, a microtubule depolymerase, is directly involved in DSB repair, contributing to the DNA damage-induced motion of DSBs. The authors identify that Kif2C binds to linear but not circular DNA in *Xenopus* and HeLa cell extracts (Figure 1). They show that Kif2C localizes to laser-induced DNA damage within 1 minute (Figure 2). The initial recruitment is dependent on PARP activity and stable association depends on ATM activity. They were able to map two critical sites in the N-terminal region of Kif2C that are both required for association with laser damage. Using siRNA knockdown and a U2OS knockout cell line the authors show that Kif2C is required to suppress accumulation of γ-H2AX, a marker for DNA damage in cycling cells without exogenous DNA damage (Figure 3). They show by back complementation that the MT depolymerase activity of Kif2C is required for this effect. Finally, the authors a small but reproducible effect on viability after cis-Pt treatment of Kif2C knockdown or knockout in multiple cell systems. The main conclusion is well supported by the data. Figure 4 expands this data into a time course analysis of γ-H2AX accumulation and turnover, showing an effect of Kif2C knockdown or knockout on γ-H2AX turnover after IR. Reporter assays show a clear effect of Kif2C knockdown on NHEJ and HR. Figure 5 demonstrates that genetic or pharmacological Kif2C turnover and fusion. Consistent with the regulation by PARP and ATM, pharmacological inhibition of PARP and ATM result in defect of 53BP1 focus mobility in wild type cell san dot a lesser degree in the Kif2C knockout cells, suggesting that these regulators target Kif2C and other proteins contributing to 53BP1 focus mobility. The conclusions are well supported by the data. Some of the data could be presented better and with more information, but overall, I do not see the need for additional experimentation.

Essential revisions:

1) Figure 1 needs additional description about the exact DNA substrates being used. The legend refers twice to the Material and Methods sections, but there is no description of the substrates or the purification of the MBP-Kif2C. This information should be added.

Please define control beads: Just empty beads or coupled to a control protein?

2) Figure 2:

I think it would help to have the co-localization of Kif2C and γ-H2AX be quantified and added to this figure. Does all Kif2C co-localize with γ H2AX? Do all γ H2AX foci co-localize with Kif2C?

Why is the ATM/ATRi not plotted in E?

3) Figure 4:

Parts A-D need quantitation.

The data is E and F are normalized. What% GFP positive cells correspond to 1? This information should be given in the legend or the% GFP-positive cells be plotted.

4) Figure 5:

For G, H, and I the time point information is missing.

Reviewer #2:

In this article, Zhu and colleagues, study the role of the kinesin Kif2C in the damage response. The authors show that Kif2C is associated with DNA repair proteins and recruited to DSBs at early repair steps. They identified the importance of ATR and PARP1 in this recruitment, and mapped Kif2C regions required for loading to damage sites. Further, they show that Kif2C defects affect both NHEJ and HR repair pathways, along with the mobility of repair sites. Kif2C is required for microtubule stability in mitosis. The data collected in this study suggest the possibility that microtubule destabilization at repair sites facilitates repair progression.

This study convincingly shows that Kif2C is recruited to repair sites, and support the model that Kif2C is a bona fide repair component. However, the data supporting a direct role of this component in repair and particularly on the nuclear dynamics of repair foci are less convincing. Importantly, the role of Kif2C microtubule-binding and depolymerizing activity in this pathway remains poorly characterized, as is the role of nuclear microtubules. Previous studies showed that cytoplasmic microtubules promote the nuclear dynamics of repair sites through the LINC complex in mammalian cells, and recent work uncovered nuclear microtubules responsible for the dynamics of repair sites in budding yeast. This paper suggests the possibility that nuclear microtubule exist in mammalian cells and their disassembly by Kif2C facilitates repair. However, this is not tested directly, leaving us wonder whether the function of Kif2C identified here is in any way linked to microtubule metabolism, or it represents an independent and unrelated role of Kif2C. Critical unanswered points include whether the nuclear function of Kif2C in repair is linked with its microtubule depolymerizing activity, and the existence of nuclear microtubules during DNA repair (particularly in the absence of Kif2C depolymerizing activity). Additionally, some experiments are missing critical controls or statistical validation, as highlighted below.

Essential revisions:

1) Kif2C physically interacts with DNA ends and repair proteins, as shown with biochemical approaches in *Xenopus* extracts and HeLa cell extracts in the presence of DNA damage. However, this interaction does not appear to be driven by DNA damage (Figure—figure supplement 1). Do the authors think Kif2C is constitutively associated with these proteins? Or is the damage dose used in these CoIP not sufficient to reveal a damage-induced interaction? Alternatively, this could indicate a cytoplasmic-Kif2C-dependent transport of these proteins, which would not support the role of this interaction in repair. It is worth testing the interaction at higher doses of damage or with different damage sources.

2) A nice analysis of domains required for recruitment to DNA damage is provided, suggesting that the N-terminal region, particularly AA86-90 and the neck domain, mediates this recruitment. However, this interpretation assumes these mutants are stable end expressed at normal levels in the nuclei. This should be shown by western blot. We note that the D86-90 mutant appears to form aggregates in the nuclei, which might reflect an unstable state of the protein. Is this a common phenotype for this mutant? Additionally, are mutated proteins still able to interact with PARP1 and ATM? If not, this would provide a mechanistic explanation of this recruitment defect.

3) Biochemical experiments in Figure 3 show that loss of Kif2C results in increased level of spontaneous damage, which is suppressed by Kif2C WT but not G495 and R420 mutants that disable Kif2C from interacting with microtubules and depolymerizing them. This is an important point that should be addressed in experiments more directly targeted to understanding the DSB repair response. Specifically, pH2AX increase could be detected in different conditions, including: (i) accumulation of cells in S-phase, (ii) accumulation of cells in mitosis (at least in some cell types); (iii) increased apoptosis; (iv) telomere dysfunction. To make the point that these mutants have impaired ability to repair DSBs, the authors should for example show that the resolution of IR-induced damage is affected (as in Figure 4). These mutants also need to be validated as in point 2) above. Specifically, the stability, expression level, nuclear enrichment, and damage recruitment of these mutant proteins should be established.

4) Similarly, for the experiments shown in Figure 3 to be valid, a control of the effect of each mutation or RNAi on the cell cycle should be provided, or effects should be tested in arrested cells (like in panel H). This is also critical for a correct interpretation of Figure 4, given that cell cycle defects can be the source of repair defects in these assays.

5) In Figure 4B,D, the sample of the UNT time point is missing, without which these experiments are uninterpretable. Are we looking here at the spontaneous damage in Kif2C-KD or the IR-induced damage? If the damage is already present in UNT conditions, this experiment has a completely different interpretation. Along these lines, it would be important to confirm these kinetics with quantifications of repair foci in fixed cells after different time points from damage induction, to circumvent many confounding effects mentioned in 3).

6) The authors propose the model that microtubule depolymerization by Kif2C is required for repair. If this is the case, taxol treatment right before and during IR should mimic the effects observed in Figure 4B,D, while Nocodazole treatment might suppress the defects observed in 4B,D. These experiments would support the authors' hypothesis.

7) Figure 5, Figure 6 show a very mild defect of taxol or Kif2C inactivation on nuclear dynamics of repair sites. Notably, these effects are much smaller than those previously observed at eroded telomeres (Lottersberger, 2015). First, the authors need to show a statistical validation for MSD analyses (not just for distance travel calculations), to demonstrate these assays are sufficiently robust in their hands to detect low effects. Second, they need to confirm the effects observed (MSD analyses, effect on distance traveled, focus clustering) are directly linked to a reduction in chromatin dynamics, rather than a general effect on cellular dynamics. Affecting cytoplasmic microtubules likely reduces cell dynamics altogether (translational movements, torsional effects, nuclear positioning), all of which would artificially reduce nuclear dynamics in tracking experiments. A necessary control for those assays is the analysis of undamaged loci in the same conditions.

8) Most importantly, are nuclear microtubules actually detected in these cells in response to damage, at least in conditions when Kif2C is inactivated?

Reviewer #3:

In this manuscript, Zhu et al., report a novel role for the Kinesin Kif2C during DNA Double Strand break repair. They report that Kif2C is recruited at DSB (using DSB mimicking oligos and laser microirradiation) in a manner that is regulated by PARP and ATM. They found that Kif2C deficient cells show enhanced γ-H2AX signaling and damage accumulation, as well as increased sensitivity to etoposide and cisplatin, and delayed γ-H2AX foci clearance following IR. Their work also indicates impaired HR and NHEJ on reporter assays. Using live imaging, they further report that Kif2C depletion decrease DSB mobility as well as DSB clustering.

Overall this work reveals interesting new insights in DSB repair biology, and provide convincing data that indeed, Kif2C contributes to DSB repair. However, the mobility and clustering section is less developed and needs to be clarified and reinforced by additional experiments.

Essential revisions:

1) Figure 5A-C: Kif2C depletion, Taxol and DHTP decrease DSB mobility. However, here MSD of 53BP1-GFP was measured, which precludes the initial condition (before damage) to be measured. It is hence difficult to conclude that Kif2C affects the mobility of the DSB and it could also impair general chromatin mobility even without damage. While I realize this is a difficult question to address, the authors could at least investigate the impact of Kif2C depletion on the mobility of other loci without etoposide treatment (telomere, centromeres etc…)

2) I have a number of concerns regarding the Figure 5F-I.

2.1) It is unclear what is shown on Figure 5E. How long after etoposide treatment is the time 0 acquired? Can we see movies? Indeed, the examples showed in circles could also be disappearance events.

2.2) It is unclear what is shown Figure 5F. If I understood well, they first treated with taxol, DHTP (or used KO Kif2C U2OS) and then with etoposide. How long after etoposide treatment were the foci counted? If this is correct, can the author change the "pre-etoposide treatment" title as this is very confusing? Moreover, I cannot see what the sentence subsection “Kif2C mediates the movement of DSBs, and the formation, fusion, and resolution of DNA damage foci” relates to. ("Because the level of DNA damage is rather elevated under Kif2C suppression, this finding indicates that the clustering of DSB is at least partially dependent on Kif2C"). In Figure 3C, DNA damage was assayed without any etoposide treatment, and here foci are counted following DSB production. So, I do not understand how they can conclude from this that DSB clustering is dependent of Kif2C and compare foci number of Figure 5 with γ-H2AX signal from Figure 3 since this is not done in the same conditions.

2.3) Somehow they imply throughout the manuscript, that foci formation is driven by DSB clustering, and that repair foci can only be detected if clustering occurs (see also in the Discussion section "Our findings indicate that […]to propel the physical movement and clustering of DSBs to enable the formation of DNA damage foci"). This is not supported by previous work, since both γ-H2AX and 53BP1 can spread on megabase of surrounding DNA. Hence even if clustering is impaired one should still detect foci, of smaller size. So, are the etoposide induced foci detected smaller in Kif2C KO? Upon taxol and DHTP treatment?

2.4) Moreover, Kif2C depletion could simply affect 53BP1 foci formation by regulating 53BP1 binding or expression. Can the author show 53BP1 western blot in Figure 3?

2.5) Related to former points, the authors need to show the images as well as quantification of the number and size of both γ-H2AX and 53BP1 foci, in the different conditions both before and after etoposide treatment. This would considerably clarify the picture and strengthen their conclusions.

2.6) Figure 5H, the decrease in foci disappearance, is in good agreement with their data on the role of Kif2C in DSB repair, but does not tell much about the dynamics of foci in the nucleus.

2.7) Finally, for Figure 5I, in the Materials and methods section, it states that 2D projections were produced for Z stacks. Were foci fusion events counted on these projections? If yes, given that both the amount of cells analyzed (about 20) and the number of fusion events per cells (about 3-5) are low and that the difference between Kif2C KO and WT is rather small (only 1-2 fusion event difference in average), it is difficult to conclude here that indeed, foci fusion is decreased. Counting should be done on unstacked images and using a higher number of cells.

[Editors’ note: further revisions were suggested prior to acceptance, as described below.]

Thank you for resubmitting your work entitled "Kinesin Kif2C in regulation of DNA double strand break dynamics and repair" for further consideration by *eLife*. Your revised article has been evaluated by Kevin Struhl (Senior Editor) and a Reviewing Editor.

The manuscript has been improved but there are some remaining issues that need to be addressed before acceptance, as outlined below:

Essential revisions:

In their revised manuscript, Zhu et al., have carefully addressed almost all previous concerns, and have improved substantially the manuscript. Live imaging with CenpB and PRPF6 provides excellent controls for their 53BP1-GFP mobility analyses, and the text has been reshaped to avoid confusion or misleading statements. This makes the study much more compelling. A few unanswered points remain, which will be restated here, and which should be addressed by text changes/additions (Points #1-3) and added analysis (#4) before publication in *eLife*.

1) In Figure 4 G and H the NHEJ and HR efficiency is normalized to 1 in wild type. Please provide the% GFP-positive cells that correspond to this value in the figure legend that the reader has an appreciation of the experimental quality of the assay.

2) Aymard et al., 2017, should be cited when mentioning the LINC complex, as in this paper, the LINC was shown to promotes DSB clustering.

3) All the experiments with Kif2C G495A and R420S mutants do not directly test the effects on damage repair. They show that in the absence of these components there is an increase in accumulation of spontaneous damage, which could be an indirect effect. To test a role in damage repair, damage should be induced, and repair kinetics should be followed (e.g., by looking at repair foci, or using a comet assay). Showing the accumulation of spontaneous damage does not address the question.

Directly related to this point, in the absence of a direct measurement of repair efficiency the text includes several overstatements. These should be toned down. Among the problematic statements are:

"…our studies using established Kif2C mutants and inhibitor supported the involvement of the ATPase and tubulin-binding activities of Kif2C in DNA repair".

"the characterization of Kif2C as a new DDR factor that mediates DNA damage movement and foci formation, in a manner involving its MT depolymerase activity, sheds new light on the spatiotemporal regulation of DNA damage dynamics"

"Together, these findings demonstrated that both the DNA damage recruitment of Kif2C and its catalytic activity are involved in the DDR."

4) Figure 2 A quantification for Figure 2I (similar to B,E,G) is still missing. Additionally, a quantification over three independent replicates should be provided to account for variability across experiments, and the authors have the data available. Why not provide it?

---

## [Author Response]

[Editors’ note: the authors resubmitted a revised version of the paper for consideration. What follows is the authors’ response to the first round of review.]

Reviewer #1:Essential revisions:1) Figure 1 needs additional description about the exact DNA substrates being used. The legend refers twice to the Material and Methods sections, but there is no description of the substrates or the purification of the MBP-Kif2C. This information should be added.Please define control beads: Just empty beads or coupled to a control protein?

As suggested, more detailed experiment information is provided in Materials and methods section for DNA substrates and MBP-Kif2C, in the “DNA binding assay” and “Pull-down assay” sections. Control pull-downs with blank beads were clarified where they were used.

2) Figure 2:I think it would help to have the co-localization of Kif2C and γ-H2AX be quantified and added to this figure. Does all Kif2C co-localize with γ H2AX? Do all γ H2AX foci co-localize with Kif2C?Why is the ATM/ATRi not plotted in E?

As shown in Figure 2C and Figure 2—figure supplement 2A, almost all γ-H2AX foci contain Kif2C, but only a portion of Kif2C co-localizes with γ-H2AX. In panel E, ATM/ATRi (caffeine) is not plotted because its effect is similar to ATMi. Due to this redundancy and the size of the panel, we did not include caffeine in the panel. However, we now mention this similar effect of caffeine in the figure legend. Moreover, we improved the Kif2C DNA damage foci study in the revised manuscript, as described below:

a) We included an immunofluorescence control in Kif2C KO cells (Figure 2—figure supplement 2B).

b) High-resolution imaging of Kif2C and 53BP1 foci co-localization is shown in Figure 2—figure supplement 1 and Video 2.

c) The co-movement of Kif2C and 53BP1 foci was analyzed in Figure 2—figure supplement 1. This new result confirms the co-localization and association between Kif2C and 53BP1 foci.

d) Statistical analyses are added to quantifications in Figure 2E and 2G.

3) Figure 4:Parts A-D need quantitation.The data is E and F are normalized. What% GFP positive cells correspond to 1? This information should be given in the legend or the% GFP-positive cells be plotted.

As suggested, quantification is provided for Figure 4A-D (γ-H2AX kinetics). The repair assay is better explained in the figure legend—GFP was measured by immunoblotting (normalized to β-actin).

4) Figure 5:For G, H, and I the time point information is missing.

Time point information is now included in the legend and Materials and methods section.

Reviewer #2:[…] This paper suggests the possibility that nuclear microtubule exist in mammalian cells and their disassembly by Kif2C facilitates repair. However, this is not tested directly, leaving us wonder whether the function of Kif2C identified here is in any way linked to microtubule metabolism, or it represents an independent and unrelated role of Kif2C. Critical unanswered points include whether the nuclear function of Kif2C in repair is linked with its microtubule depolymerizing activity, and the existence of nuclear microtubules during DNA repair (particularly in the absence of Kif2C depolymerizing activity). Additionally, some experiments are missing critical controls or statistical validation, as highlighted below.

We thank the reviewer for the insightful comments. As suggested by the reviewer, our revised manuscript provides additional evidence to better characterize the role of Kif2C in DNA repair and DNA damage foci dynamics. Detailed responses to these comments are provided below:

Essential revisions:1) Kif2C physically interacts with DNA ends and repair proteins, as shown with biochemical approaches in *Xenopus* extracts and HeLa cell extracts in the presence of DNA damage. However, this interaction does not appear to be driven by DNA damage (Figure S1). Do the authors think Kif2C is constitutively associated with these proteins? Or is the damage dose used in these CoIP not sufficient to reveal a damage-induced interaction? Alternatively, this could indicate a cytoplasmic-Kif2C-dependent transport of these proteins, which would not support the role of this interaction in repair. It is worth testing the interaction at higher doses of damage or with different damage sources.

We confirmed that the damage condition used was sufficient in inducing DNA damage signaling (Figure 1—figure supplement 1B). This does indicate that a portion of Kif2C constitutively associates with DNA repair factors. This pattern is common for DNA damage response factors which often exhibit constitutive associations even without DNA damage. Thus, this association between Kif2C with repair factors further suggests a role of Kif2C in DNA repair. The alternative possibility to explain these associations, as mentioned by the reviewer, was that Kif2C mediates the transport of these repair factors. However, unlike most Kif proteins, Kif2C does not transport cargos, and is itself predominantly a nuclear protein in interphase. Our studies subsequently revealed the DNA damage recruitment of Kif2C and its functional importance for DNA repair and foci movement, all supporting a direct function of Kif2C in DNA repair.

2) A nice analysis of domains required for recruitment to DNA damage is provided, suggesting that the N-terminal region, particularly AA86-90 and the neck domain, mediates this recruitment. However, this interpretation assumes these mutants are stable end expressed at normal levels in the nuclei. This should be shown by western blot. We note that the D86-90 mutant appears to form aggregates in the nuclei, which might reflect an unstable state of the protein. Is this a common phenotype for this mutant? Additionally, are mutated proteins still able to interact with PARP1 and ATM? If not, this would provide a mechanistic explanation of this recruitment defect.

We confirmed that Kif2C mutants were expressed at similar levels as WT Kif2C (Figure 3F and Figure 2—figure supplement 3). Like WT Kif2C, these mutants were localized predominantly in the interphase nucleus (Figure 2I). We did notice some punctate foci for Δ86-90 and other mutants (WT to lesser extent), which may indicate aggregation. However, the majority of all proteins exhibited similar, and predominantly pan-nuclear expression. As suggested by the reviewer, we examined the association of WT, Δ86-90 and Δneck Kif2C with PARP1, Ku70, and γ-H2AX. The mutants which were deficient in DNA damage recruitment also showed reduced associations with DNA repair proteins (Figure 2—figure supplement 3).

3) Biochemical experiments in Figure 3 show that loss of Kif2C results in increased level of spontaneous damage, which is suppressed by Kif2C WT but not G495 and R420 mutants that disable Kif2C from interacting with microtubules and depolymerizing them. This is an important point that should be addressed in experiments more directly targeted to understanding the DSB repair response. Specifically, pH2AX increase could be detected in different conditions, including: (i) accumulation of cells in S-phase, (ii) accumulation of cells in mitosis (at least in some cell types); (iii) increased apoptosis; (iv) telomere dysfunction. To make the point that these mutants have impaired ability to repair DSBs, the authors should for example show that the resolution of IR-induced damage is affected (as in Figure 4). These mutants also need to be validated as in point 2) above. Specifically, the stability, expression level, nuclear enrichment, and damage recruitment of these mutant proteins should be established.

This portion of study is improved as follow:

a) The accumulation of DNA damage due to Kif2C depletion is now also confirmed by foci formation of γ-H2AX and 53BP1, which was rescued by WT but not mutant Kif2C (Figure 3—figure supplement 1).

b) Comet assay which directly indicates DNA breaks confirmed the elevated DNA damage and reduce repair in Kif2C KO cells (Figure 4—figure supplement 1).

c) The expression level, nuclear localization, and DNA damage recruitment of these mutants are established in Figure 3D, 3E and Figure 3—figure supplement 2A.

d) The above points indicate direct function of Kif2C in DNA repair, in comparison to indirect effects of cell cycle defects. To further address this concern, we showed that Kif2C depletion did not significantly alter cell cycle progression (Figure 3—figure supplement 2B); this is consistent with the normal viability of KO cells; neither Kif2C KO nor mutant expression exhibited cell cycle deficiencies (Figure 3—figure supplement 2B).

e) The manuscript was edited to include the above points.

4) Similarly, for the experiments shown in Figure 3 to be valid, a control of the effect of each mutation or RNAi on the cell cycle should be provided, or effects should be tested in arrested cells (like in panel H). This is also critical for a correct interpretation of Figure 4, given that cell cycle defects can be the source of repair defects in these assays.

As suggested by the reviewer, the concern of cell cycle deficiency is addressed by FACS, neither Kif2C KO or mutant expression exhibited cell cycle deficiencies (Figure 3—figure supplement 2B).

5) In Figure 4B,D, the sample of the UNT time point is missing, without which these experiments are uninterpretable. Are we looking here at the spontaneous damage in Kif2C-KD or the IR-induced damage? If the damage is already present in UNT conditions, this experiment has a completely different interpretation. Along these lines, it would be important to confirm these kinetics with quantifications of repair foci in fixed cells after different time points from damage induction, to circumvent many confounding effects mentioned in 3).

As suggested, the untreated time points are included. Although Kif2C depletion led to higher endogenous DNA damage, the kinetics of the induction and repair of DNA damage after IR are clearly shown. Quantification is also provided to illustrate the kinetics (Figure 4C and 4F). Consistent result was also provided by the comet assay measuring DNA breaks (Figure 4—figure supplement 1).

6) The authors propose the model that microtubule depolymerization by Kif2C is required for repair. If this is the case, taxol treatment right before and during IR should mimic the effects observed in Figure 4B,D, while Nocodazole treatment might suppress the defects observed in 4B,D. These experiments would support the authors' hypothesis.

Unlike Kif2C depletion, taxol or nocodazole exhibits strong toxicity by impacting all cellular processes involving microtubules. For example, these drugs will lead to substantial mitotic arrest and cell death during the 6-hour treatment, as needed for repair kinetics in Figure 4. These confounding effects of taxol/nocodazole make measuring DNA repair challenging, and results difficult to interpret. However, along with the reviewer’s point, we were able to confirm that, like Kif2C suppression, taxol reduced DNA damage mobility and foci dynamics (Figure 5). These experiments were carried out in a much shorter time window (10-20 minutes), in morphologically normal interphase cells.

7) Figure 5, Figure 6 show a very mild defect of taxol or Kif2C inactivation on nuclear dynamics of repair sites. Notably, these effects are much smaller than those previously observed at eroded telomeres (Lottersberger, 2015). First, the authors need to show a statistical validation for MSD analyses (not just for distance travel calculations), to demonstrate these assays are sufficiently robust in their hands to detect low effects. Second, they need to confirm the effects observed (MSD analyses, effect on distance traveled, focus clustering) are directly linked to a reduction in chromatin dynamics, rather than a general effect on cellular dynamics. Affecting cytoplasmic microtubules likely reduces cell dynamics altogether (translational movements, torsional effects, nuclear positioning), all of which would artificially reduce nuclear dynamics in tracking experiments. A necessary control for those assays is the analysis of undamaged loci in the same conditions.

We have made significant improvement in this part of the manuscript, as explained below:

a) Our mean square displacement (MSD) data are in fact quite consistent with those reported by Lottersberger et al., 2015. Our MSD value at 10 minutes in control condition is around 0.65 μm^2^ (Figure 5B and 5C), consistent with theirs at ~0.6 μm^2^. For the effect of taxol, Lottersberger et al., was 20 μM, which is 4 times higher than the dose we used in our study. We did try higher concentrations of taxol in our pilot assays and obtained similar value as theirs. At 10 μM taxol, our MSD value at 10 minutes is at around 0.2 μm^2^ (Author response image 1), similar to their reported value at ~0.2 μm^2^ with 20 μM taxol. To reveal more specific effects and avoid excessive general toxicity of taxol (such as the reduction of general nuclear dynamics shown in Figure 5—figure supplement 2), we chose this lower dose for this study.

**Author response image 1. respfig1:** Mean ssquare displacement analysis of EGFP-53BP1 monility over 10 minutes in WT-U2OS cells DMSO control and in the presence of 5 μM or 10 μM taxol.

b) As in Lottersberger et al., 2015, statistics are typically not applied to MSD analyses because errors amplify with increasing time intervals (“square” of displacement in Brownian motion). For this reason, we resorted to perform statistical analysis on the distance travelled in 10 minutes as in Lottersberger et al., 2015, shown in Figure 5D, to distinguish the effect of Kif2C KO or inhibition from that of WT. With that being said, we are open to better suggestions.

c) As the most significant addition, we have now included two controls: CenpB (for undamaged, and general chromatin dynamics) and PRPF6 (for general intra-nuclear dynamics). Both CenpB and PRPF6 form distinct punctate foci in the nucleus that are not related to DNA damage. Interestingly, intra-nuclear CenpB and PRPF6 foci are relatively less dynamics than 53BP1 foci, and there was no significant difference between WT and Kif2C KO or DHTP treatment (Figure 5—figure supplement 1). Remarkably, taxol treatment (even at 5 µM) reduced all foci dynamics (Figure 5—figure supplement 2). Based on these data, we concluded that Kif2C is a specific regulator in DDR and DHTP treatment is much more selective than MT poisons, with no effect on other cellular dynamics in general. We believe these controls provide new and important information for not only our current study, but also previous studies defining the mobility of damaged chromatin.

8) Most importantly, are nuclear microtubules actually detected in these cells in response to damage, at least in conditions when Kif2C is inactivated?

Although our data may suggest the existence of nuclear microtubules, we understand this is a provocative point. As discussed in the manuscript, DNA damage-induced intra-nuclear microtubule filaments were visualized in yeast cells, but not yet in mammalian cells. We carefully discuss all possibilities, for example, nuclear MTs may not exist as long hollow tubules like cytoplasmic MTs. This possibility reminded us about the early days of proving the existence of nuclear F-actin. On the other hand, instead of forming MT filaments, dynamic tubulin assembly/disassembly, mediated by Kif2C and other MT regulators, may participate in DNA repair. We feel that visualization of nuclear MT is beyond the scope of this manuscript and will require a much more substantial and focused effort in future studies.

Reviewer #3:[…] Overall this work reveals interesting new insights in DSB repair biology, and provide convincing data that indeed, Kif2C contributes to DSB repair. However, the mobility and clustering section is less developed and needs to be clarified and reinforced by additional experiments.

We thank the reviewer for the insightful comments. As suggested by the reviewer, the mobility and clustering section is now better developed, as explained below:

Essential revisions:1) Figure 5A-C: Kif2C depletion, Taxol and DHTP decrease DSB mobility. However, here MSD of 53BP1-GFP was measured, which precludes the initial condition (before damage) to be measured. It is hence difficult to conclude that Kif2C affects the mobility of the DSB and it could also impair general chromatin mobility even without damage. While I realize this is a difficult question to address, the authors could at least investigate the impact of Kif2C depletion on the mobility of other loci without etoposide treatment (telomere, centromeres etc…)

As suggested by the reviewer, we have now included two controls: CenpB (for undamaged, and general chromatin dynamics) and PRPF6 (for general intra-nuclear dynamics). Both CenpB and PRPF6 form distinct punctate foci in the nucleus that are not DNA damage-related, and these control experiments turn out quite interesting. Intra-nuclear CenpB and PRPF6 foci are relatively less dynamics than 53BP1 foci, and there was no significant difference between WT and Kif2C KO or DHTP treatment (Figure 5—figure supplement 1). On the other hand, Taxol treatment reduced all foci dynamics (Figure 5—figure supplement 2). Based on these data, we concluded that Kif2C is a specific regulator in DDR and DHTP treatment is much more selective than MT poisons, with no effect on other cellular dynamics in general. We believe these controls provide new and important information for not only our current study, but also previous studies defining the mobility of damaged chromatin.

2) I have a number of concerns regarding the Figure 5F-I.2.1) It is unclear what is shown on Figure 5E. How long after etoposide treatment is the time 0 acquired? Can we see movies? Indeed, the examples showed in circles could also be disappearance events.

As better explained now in the revised manuscript, time 0 is 5 minutes after etoposide treatment (for microscopy set up time and for consistency). We have also included representative movies to illustrate these foci events (Video 3 and Video 4). Indeed, as shown in these movies, the fusion and disappearance events are quite distinct from each other. For fusion events, the two foci move across and get close to each other right before fusion, and we observed a brighter focus formed as a result of the two fused foci.

2.2) It is unclear what is shown Fig5F. If I understood well, they first treated with taxol, DHTP (or used KO Kif2C U2OS) and then with etoposide. How long after etoposide treatment were the foci counted? If this is correct, can the author change the "pre-etoposide treatment" title as this is very confusing? Moreover, I cannot see what the sentence subsection “Kif2C mediates the movement of DSBs, and the formation, fusion, and resolution of DNA damage foci” relates to. ("Because the level of DNA damage is rather elevated under Kif2C suppression, this finding indicates that the clustering of DSB is at least partially dependent on Kif2C"). In Figure 3C, DNA damage was assayed without any etoposide treatment, and here foci are counted following DSB production. So, I do not understand how they can conclude from this that DSB clustering is dependent of Kif2C and compare foci number of Figure 5 with γ-H2AX signal from Figure 3 since this is not done in the same conditions.

In this experiment, taxol or DHTP treatment was done 5 minutes before etoposide treatment (therefore was called “pre-etoposide treatment”). We have rephrased it to “pre-treatment (before etoposide)”. The foci were counted 5 minutes after etoposide treatment. These descriptions are now better explained in the legend.

As suggested by the reviewer, new results are added to confirm that Kif2C suppression increased the level of DNA damage induced by etoposide (Figure 4—figure supplement 2C). Relative statements in the manuscript were also rephrased to avoid potential overstatement or confusion.

2.3) Somehow they imply throughout the manuscript, that foci formation is driven by DSB clustering, and that repair foci can only be detected if clustering occurs (see also in the Discussion section "Our findings indicate that […]to propel the physical movement and clustering of DSBs to enable the formation of DNA damage foci"). This is not supported by previous work, since both γ-H2AX and 53BP1 can spread on megabase of surrounding DNA. Hence even if clustering is impaired one should still detect foci, of smaller size. So, are the etoposide induced foci detected smaller in Kif2C KO? Upon taxol and DHTP treatment?

We agree with the reviewer that γ-H2AX and 53BP1 focus can be induced by a single DSB (and potentially detectable at a higher resolution). However, previous evidence demonstrated that the macro-domains commonly defined as DNA damage foci involved clustering of multiple DNA damage sites (Neumaier et al., 2012; Aten et al., 2004; Aymard et al., 2017; Roukos et al., 2013; Asaithamby and Chen,2011). Our initial reasoning was that this clustering process must rely on DNA damage mobility. In light of the reviewer’s suggestion, we rephrased the statement. We withdrew the notion that foci formation is enabled by DNA damage movement and clustering. Instead, we showed that Kif2C mediated both DSB mobility and foci formation/fusion, which provided new evidence to potentially connect these processes at the mechanistic level.

2.4) Moreover, Kif2C depletion could simply affect 53BP1 foci formation by regulating 53BP1 binding or expression. Can the author show 53BP1 western blot in Figure 3?

We confirmed in the revision that Kif2C depletion did not alter 53BP1 expression (Figure 4—figure supplement 2B).

2.5) Related to former points, the authors need to show the images as well as quantification of the number and size of both γ-H2AX and 53BP1 foci, in the different conditions both before and after etoposide treatment. This would considerably clarify the picture and strengthen their conclusions.

As discussed above, we have re-justified conclusions related to foci formation and DNA damage mobility. As for foci measurement in conditions before and after etoposide, we observed 53BP1 foci formation in our live-cell imaging only after etoposide treatment.

2.6) Figure 5H, the decrease in foci disappearance, is in good agreement with their data on the role of Kif2C in DSB repair, but does not tell much about the dynamics of foci in the nucleus.

The following improvements were made in the foci disappearance study:

a) The conclusion is more carefully justified. We agree with the reviewer that foci disappearance may reflect repair, we do not rule out the possibility of foci disassembly (resolution), e.g., one focus is resolved and the released, unrepaired DSBs join other foci.

b) A high-resolution movie is added to provide an example of foci disappearance (Video 4).

c) To better illustrate the dynamics of foci disappearance, we have added a new panel showing the time taken for the foci to disappear under each condition (Figure 5J). The time of disappearance was longer in Kif2C KO cells than in the WT, and both taxol and DHTP treatment lengthened the time of disappearance. Most foci disappeared in the first 30 minutes in the WT cells, drastically differed from those under other experimental conditions (Kif2C KO, taxol and DHTP, Figure 5I).

2.7) Finally, for Figure 5I, in the Materials and methods section, it states that 2D projections were produced for Z stacks. Were foci fusion events counted on these projections? If yes, given that both the amount of cells analyzed (about 20) and the number of fusion events per cells (about 3-5) are low and that the difference between Kif2C KO and WT is rather small (only 1-2 fusion event difference in average), it is difficult to conclude here that indeed, foci fusion is decreased. Counting should be done on unstacked images and using a higher number of cells.

We apologize for the confusion: the 2D projections were used only for presentations in the figures and the related videos. For the tracking of the foci and counting the disappearance and fusion events, we did not use the projection from the Z stacks. We identified foci in different Z stacks throughout the time series to determine the occurrence of disappearance and fusion events. We also used this positional tracking for our analysis of DSB mobility. To clarify this, we added more details about the tracking and data analyses to the methods section in the revised manuscript. With regard to low number of fusion events, it should be noted that the fusion events are infrequent (even in the WT cells) and these numbers were reported as per cell. To avoid this confusion, now we present these data as total number of events in 15 cells per condition over three independent experiments which achieved clear statistical significance (Figure 5H).

[Editors’ note: what follows is the authors’ response to the second round of review.]

The manuscript has been improved but there are some remaining issues that need to be addressed before acceptance, as outlined below:Essential revisions:[...] A few unanswered points remain, which will be restated here, and which should be addressed by text changes/additions (Points #1-3) and added analysis (#4) before publication in eLife.

As suggested, points #1-3 are now addressed by corresponding text changes/additions; quantification is included in Figure 2I, per point #3.

1) In Figure 4 G and H the NHEJ and HR efficiency is normalized to 1 in wild type. Please provide the% GFP-positive cells that correspond to this value in the figure legend that the reader has an appreciation of the experimental quality of the assay.

The expression of GFP (and control actin) was measured by immunoblotting. We prefer immunoblotting over fluorescence measurement because immunoblotting is more accurate, less affected by pseudofluorescence (e.g., from cell debris), and reflects the entire cell population. At the validation stage of these experiments, we did measure the% cells with GFP-positive. The values were typically in the range of 3-10%, which are well in line with the original studies which characterized these systems. We have now included these explanations in the figure legend and methods.

2) Aymard et al., 2017, should be cited when mentioning the LINC complex, as in this paper, the LINC was shown to promotes DSB clustering.

This citation is added.

3) All the experiments with Kif2C G495A and R420S mutants do not directly test the effects on damage repair. They show that in the absence of these components there is an increase in accumulation of spontaneous damage, which could be an indirect effect. To test a role in damage repair, damage should be induced, and repair kinetics should be followed (e.g., by looking at repair foci, or using a comet assay). Showing the accumulation of spontaneous damage does not address the question.Directly related to this point, in the absence of a direct measurement of repair efficiency the text includes several overstatements. These should be toned down. Among the problematic statements are:"…our studies using established Kif2C mutants and inhibitor supported the involvement of the ATPase and tubulin-binding activities of Kif2C in DNA repair"."the characterization of Kif2C as a new DDR factor that mediates DNA damage movement and foci formation, in a manner involving its MT depolymerase activity, sheds new light on the spatiotemporal regulation of DNA damage dynamics""Together, these findings demonstrated that both the DNA damage recruitment of Kif2C and its catalytic activity are involved in the DDR."

As suggested, these statements are rephrased for improved accuracy, as below:

“our studies using established Kif2C mutants and inhibitor suggested that the ATPase and tubulin-binding activities of Kif2C were indispensable for suppressing DNA damage accumulation.”

“the characterization of Kif2C as a new DDR factor that mediates DNA damage movement and foci formation sheds new light on the spatiotemporal regulation of DNA damage dynamics.”

“these findings suggested that both the DNA damage recruitment of Kif2C and its catalytic activity are likely involved in the DDR.”

4) Figure 2 A quantification for Figure 2I (similar to B,E,G) is still missing. Additionally, a quantification over three independent replicates should be provided to account for variability across experiments, and the authors have the data available. Why not provide it?

As suggested, quantification is added for Figure 2I.